# Canadian forest fires, Icelandic volcanoes and increased local dust observed in 6 shallow Greenland firn cores

Helle Astrid Kjær[1], Patrick Zens[1], Samuel Black[1,2], Kasper Holst Lund[1], Anders Svensson[1] and Paul Vallelonga[1,3]

1 Physics for Ice, Climate and Earth Sciences (PICE), Niels Bohr Institute, University of Copenhagen, Copenhagen 2100, Denmark
2 NatureScot, Inverness, IV3 8NW, United Kingdom
UWA Oceans Institute, University of Western Australia, Crawley, WA, Australia

*Correspondence to*: Helle Astrid Kjær (Hellek@fys.ku.dk)

**Abstract.**

Greenland ice cores provide information about past climate. Few impurity records covering the past two decades exist from Greenland. Here we present results from six firn cores obtained during a 426 km long northern Greenland traverse made in 2015 between the NEEM and the EGRIP deep drilling stations situated on the Western and Eastern side of the Greenland ice sheet, respectively. The cores (9 to 14 m long) are analysed for chemical impurities and cover time spans of 18 to 53 years (±4 yrs) depending on local snow accumulation that decreases from west to east.

The high temporal resolution allows for annual layers and seasons to be resolved. Insoluble dust, ammonium, and calcium concentrations in the 6 firn cores overlap, and also the seasonal cycles are similar in timing and magnitude across sites, while peroxide ($H_2O_2$) varies spatially because it is accumulation dependent and conductivity likely influenced by sea salts, also vary spatially.

Overall, we determine a rather constant dust flux over the period, but in the recent years (1998-2015) we identify an increase in large dust particles that we ascribe to an activation of local Greenland sources. We observe an expected increase in acidity and conductivity in the mid 1970's as a result of anthropogenic emissions followed by a decrease due to mitigation. Several volcanic horizons identified in the conductivity and acidity records can be associated with eruptions in Iceland and in the Barents Sea region. From a composite ammonium record we obtain a robust forest fire proxy associated primarily with Canadian forest fires (R=0.49).

## 1 Introduction

The accumulation and preservation of past snowfall as glacier ice stores an abundance of information regarding past environmental conditions that can be retrieved through physical and chemical analyses of polar ice cores.

For Greenland, water isotopes and deuterium excess, can provide information on average temperatures and ice volumes (Johnsen et al., 1989; Dansgaard, 1964); dust layers provide constrains on large-scale atmospheric circulation patterns and

desertification (Fischer et al., 2007; Ruth et al., 2002; Marius Folden Simonsen et al., 2019); sea salts (eg. $Na^+$) further constrain atmospheric transport, while simultaneously informing on oceanic conditions (Schüpbach et al., 2018; Fischer et al., 2007; Rhodes et al., 2018); ammonium concentration maxima provide evidence of forest fire activity and global vegetation coverage (*e.g.* Legrand et al., 1992; Legrand et al., 2016). Often these proxies exhibit annual cycles in the composition and concentration

due to natural cycles in their atmospheric concentration but also as a result of temperature, accumulation and wind fluctuations at the deposition site. These annual cycles can be used to identify corresponding annual layers in the ice important for dating the high-resolution climatic signals (Rasmussen et al., 2013; Svensson et al., 2008). As an example the production of peroxide ($H_2O_2$) mainly takes place during months of intense insolation as it is produced by a photochemically derived self-reaction of hydroperoxyl radicals ($HO_2$)(Frey et al., 2006; Sigg and Neftel, 1988). Therefore, $H_2O_2$ records show maximum concentrations

in the summer and minimum concentrations during the winter months, when photochemical processes are absent at polar latitudes (Sigg and Neftel, 1988; Frey et al., 2006). However, $H_2O_2$ maintain a constant exchange with the atmosphere, leading to post-depositional relocation within the upper snow and firn. Thus if snow accumulation rates are not high enough (0.13 m w.eq. $a^{-1}$) this exchange can cause smoothing and loss of seasonal $H_2O_2$ signal will occur (Neftel, 1996).

The development of high resolution continuous flow analysis (CFA) techniques (Kaufmann et al., 2008; Bigler et al., 2011;

Dallmayr et al., 2016; Kjær et al., 2021a) has allowed obtaining continuous long-term paleoclimate records back through time on a sub-annual scale (Schüpbach et al., 2018; Marius Folden Simonsen et al., 2019) and is now a standard in ice core analyses. CFA represents a highly efficient and rapid analysis technique relative to the measurement of discrete samples, despite its intrinsic dispersion of the signal and small sample loss around core breaks and is favoured for the effective sample decontamination and high sampling resolution (Breton et al., 2012; Erhardt et al., 2022).

We evaluate the impurity concentrations as determined by means of CFA in six shallow Northern Greenland firn cores across Northern Greenland sites. The cores are dated individually to allow comparisons of temporal and spatial trends in both mean concentrations and seasonal cycles. Further we investigate extreme events, such as the deposition from forest fires and volcanic eruptions, and their representation between the 6 sites. The sites chosen cover the lower accumulation area in the central North Greenland, both east and west of the divide, and has only limited prior analysis of this kind (Du et al., 2019a; Vallelonga et

al., 2014; Fischer et al., 1998; Gfeller et al., 2014; Schüpbach et al., 2018; Kjær et al., 2021a).

## 2 Methods

Six shallow firn cores were collected during the NEEM to EastGRIP (N2E) traverse in May to June 2015 (Karlsson et al., 2020). The N2E traverse went from the NEEM (The North Greenland Eemian Ice Drilling) deep ice core drill site (77.5°N, 51.0°W, 2481 m a.s.l.) to the EastGRIP (The East Greenland Ice-core Project) deep ice core drill site (75.64°N, 36.0°W,

2712 m a.s.l.). Cores were drilled from the surface to a depth between 9.08 m and 14.02 m. The position and time period covered by the firn cores labelled T2015-A1 to T2015-A6 can be found in Table 1 and Figure 1 (Kjær et al., 2021c).

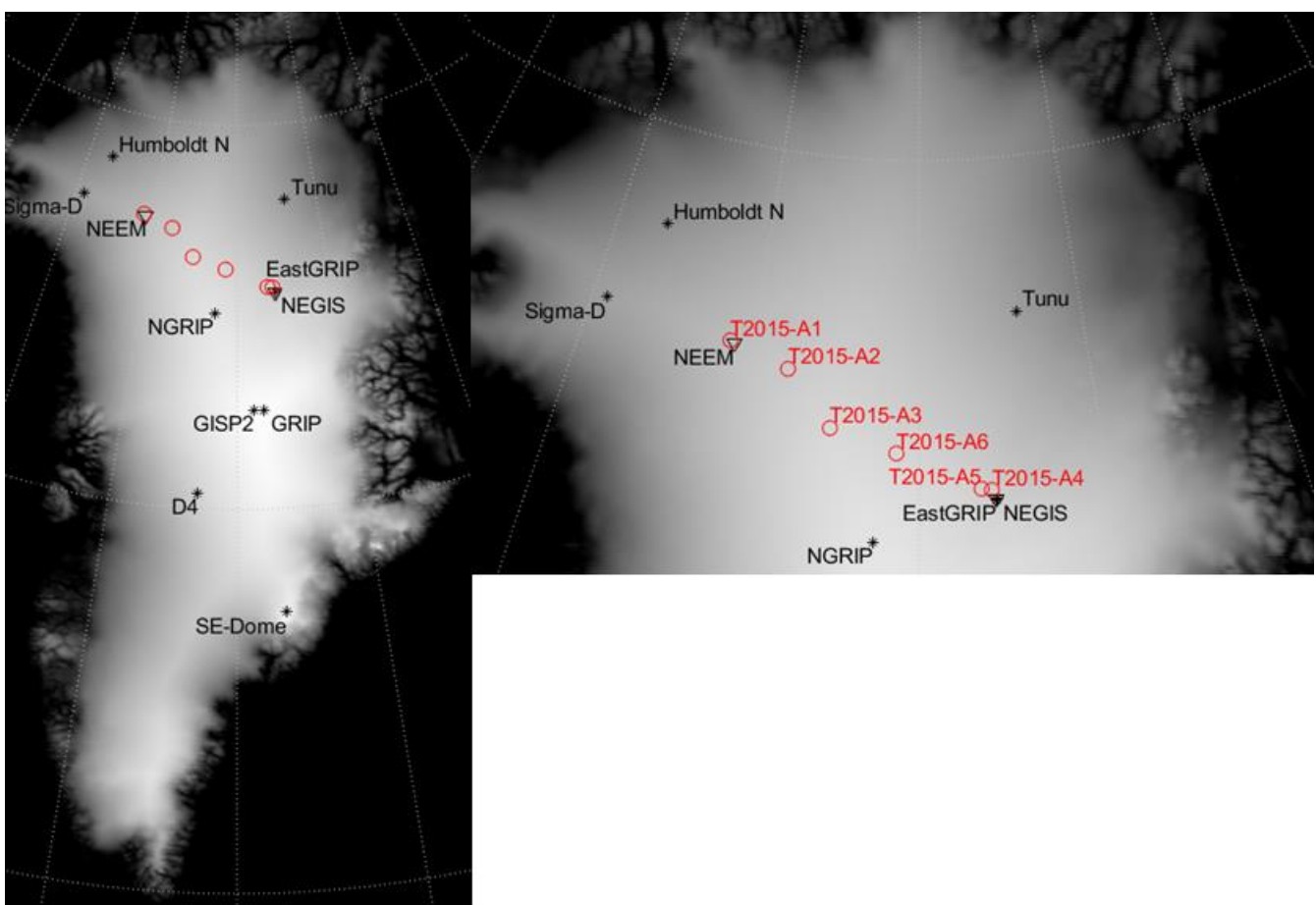

**Figure 1: The 6 drill sites for the shallow firn cores investigated in this study (red circles) as well as the sites of other ice and firn cores mentioned in this study (black stars) on a background of Greenland surface elevation (SeaRISE dataset, Bamber, 2001).**

**Table 1. Name, latitude, longitude, altitude, core depths, bottom age and mean accumulation (Kjær et al., 2021c) of the shallow firn cores presented in this study. The firn cores are labelled T2015-A1 to T2015-A6. All cores were drilled in 2015 and measured by means of CFA in Copenhagen in 2017.**

| Firn core | Coordinates | | Altitude | Depth of core | Time period covered | Mean annual accumulation |
|---|---|---|---|---|---|---|
| ID | N | W | m a.s.l. | m | AD | cm w.eq a⁻ |
| T2015-A1 (NEEM) | 77N25.219' | 51W09.588' | 2484 | 9.08 | 2015-1997 (±1) | 23.1±5.6 |
| T2015-A2 | 77N01.764' | 47W28.832' | 2620 | 10.74 | 2015-1988 (±1) | 19.6±4.6 |
| T2015-A3 | 76N27.290' | 44W47.709' | 2771 | 10.97 | 2015-1988 (±1) | 17.8±3.0 |
| T2015-A4 | 75N41.340' | 36W28.926' | 2701 | 10.91 | 2015-1980 (±2) | 14.4±3.2 |
| T2015-A5 (EGRIP) | 75N37.501' | 35W58.809' | 2708 | 14.02 | 2015-1962 (±2) | 13.6±3.6 |
| T2015-A6 | 76N10.294' | 41W05.628' | 2760 | 12.07 | 2015-1962 (±3) | 12.2±2.87 |

The firn cores were drilled using the American IDDO (U.S. Ice Drilling and Design Operations) hand auger (76 mm diameter). In the field the cores were split into 55 cm long segments and packed into plastic bags. They were transported in cooler boxes on sledges to the EastGRIP site, from where they were flown first to Kangerlussuaq, Greenland, and then shipped further to Copenhagen, Denmark for analysis. In Copenhagen they were stored at -20 ℃ until further cutting into sections of 3.4 x 3.4 cm just prior to the CFA measurements (Bigler et al. 2011).

## 2.1 Continuous flow analysis (CFA)

In 2017, two years after retrieval, the CFA system at the Niels Bohr Institute in Copenhagen (Bigler et al., 2011) was used to analyse the 6 firn cores for their chemical impurity content. The system was slightly adapted from the published Copenhagen CFA system (Bigler et al., 2011), which determines conductivity ($\sigma$), insoluble dust, ammonium ($NH_4^+$) and calcium ($Ca^{2+}$), by adding analysis of hydrogen peroxide ($H_2O_2$) and acidity ($H^+_{dye}$ Kjær et al., 2016). The flow chart and additional information for the particular CFA setup and instrumentation is presented in the supplementary material (Figure S1, Tables S1 to S3). The determined signals were converted into units of concentration using a linear regression produced by a set of two ($H_2O_2$ and $H^+_{dye}$) or three ($NH_4^+$ and $Ca^{2+}$) known standards (Table S3). Calibrations were performed every four hours. In general the baseline was established by running ultra-pure water (milliq) water through the system for every 4.4 m of firn analysed (eight pieces each 55 cm long). However, for the top 1.65 metre where the core was fragile as a result of low density, the baseline was established in between each of the top three 55 cm sections to ensure baseline stability and avoid compression from overlying cores increasing the uncertainty on the depth registration.

Driven by capillary forces the melt water percolates from the CFA melt head into the firn core above the melt head dispersing the signal. This was mitigated by adding a metal (97 % Cu, 2,5 % Zn) coin to the melt head to limit contact between any excess meltwater on the melt head and the firn core. In addition, such excess water that could be sucked up into the firn was limited by carefully adjusting melt head temperature relative to pump speeds carrying the water away. With these modifications the level of water percolating into the firn from the melt head was limited to <1cm. Melt rate was kept at ~4 cm/min which resulted in the final depth resolution of the ions measured being <2 cm ($H^+_{dye}$, $NH^{4+}$, $H_2O_2$, $Ca^{2+}$), while for the conductivity and dust with shorter step-change response times (time it takes to go from a level of 5% to 95% of a concentration) a depth resolution of 8 mm was achieved. We note that the accumulation at the sites vary between 12 and 23 cm w.eq a$^{-1}$ and thus annual signals are resolved with the achieved resolution.

## 2.1  Core chronology

The calibrated data retrieved from the CFA is shown for the individual cores on a depth scale in the supplementary material Figure S2 to S7.We rely on the strong seasonal pattern of $H_2O_2$ (Figure S2 to S7, top) to constrain the age of the 6 shallow cores (Table 1), where we assign the summer maxima of $H_2O_2$ to solar solstice (June). At the low accumulation sites where $H_2O_2$ seasonality was not well resolved; T2015-A4, T2015-A5, and T2015-A6  the seasonality in $Ca^{2+}$ (Figure S2-S7, second topmost) was used to further constrain the firn core chronologies. Despite the fact that others of the proxies analysed also show

a strong annual cycle (see Figure 2, and also Figure 4) we stick to an age scale based on just $H_2O_2$ (or $Ca^{2+}$). This is because one of the aims of the study is to investigate the seasonal cycle between sites. In addition, we note that acid horizons are commonly used to match ages between cores. However, we have chosen not to do so, as another aim for is to investigate which of the extreme acid layers in recent time that can be used to constrain ages between sites. The total age of each core and the

uncertainty was defined as $\pm$ ½ a year for each uncertain year and can be found in Table 1. We then use the age-depth relationship from the $H_2O_2$ peaks to interpolate the depth series into a time series using a constant accumulation assumption. Accumulation from the GC network at NEEM suggests that a fairly equal summer to winter ratio (Gfeller et al., 2014) and thus we stick with a simple constant accumulation scenario (Gfeller et al., 2014; Kjær et al., 2013). We could have used re-analysis accumulation data to constrain the monthly accumulation, but even high-resolution weather re-analysis performs

poorly on the central ice sheet (Kjær et al., 2021c).

To investigate the seasonality in the proxies we first removed the five year running average and we use the term excess for the remainder. The years were split further into 12 months of equal accumulation using the formal month definition (Gfeller et al., 2014; Kjær et al., 2016). We highlight that we have not used the extremes in acidity nor $NH_4^+$ to constrain the dating between the 6 firn cores and thus each core is dated on its own individual timescale using solely annual layer counting of $H_2O_2$ and in

the case of T2015-A6, T2015-A5, T2015-A4 also $Ca^{2+}$. The firn cores span 18 to 53 years depending on local snow accumulation that decreases from west to east. The uncertainty of the age scale is estimated to be $\pm 3$ years at the base of the oldest core, but is less for the remainder of the cores.

## 3 Spatial gradients

In Figure 2 the full resolution chemical data from the CFA campaign is presented for all six firn cores combined on an age

scale, while in Figures S2 to S7 they are presented individually on a depth scale. Figure 3 and Table S4 represents the median, and 15 and 85 percentiles of the individual records, while Figure 3 shows in addition the 2.5 and 97.5% range. Pearson correlations between the sites are presented for the individual proxies in the supplementary section S3 for both annual mean records and monthly mean records.

We start by comparing the individual sites to previous analysis of this kind. At the EastGRIP site (T2015-A4 and T2015-A5)

our medians (Figure 3, Table S4) are comparable with previous measurements (Vallelonga et al. 2014; Kjær et al. 2016a; Du et al. 2019).

At the NEEM site (T2015-A1) the $NH_4^+$ median (Figure 3, Table S4) is concurrent with other shallow cores (1982-2011) having concentrations of 5.5±5.7 to 8.1±8.5 ppb $NH_4^+$ and 4.7±4.7 to 6.9 ± 5.2 ppb $Ca^{2+}$ (Gfeller et al., 2014) and is also directly comparable to that of NEEM past 2000 years (Zennaro et al., 2014), suggesting no significant recent

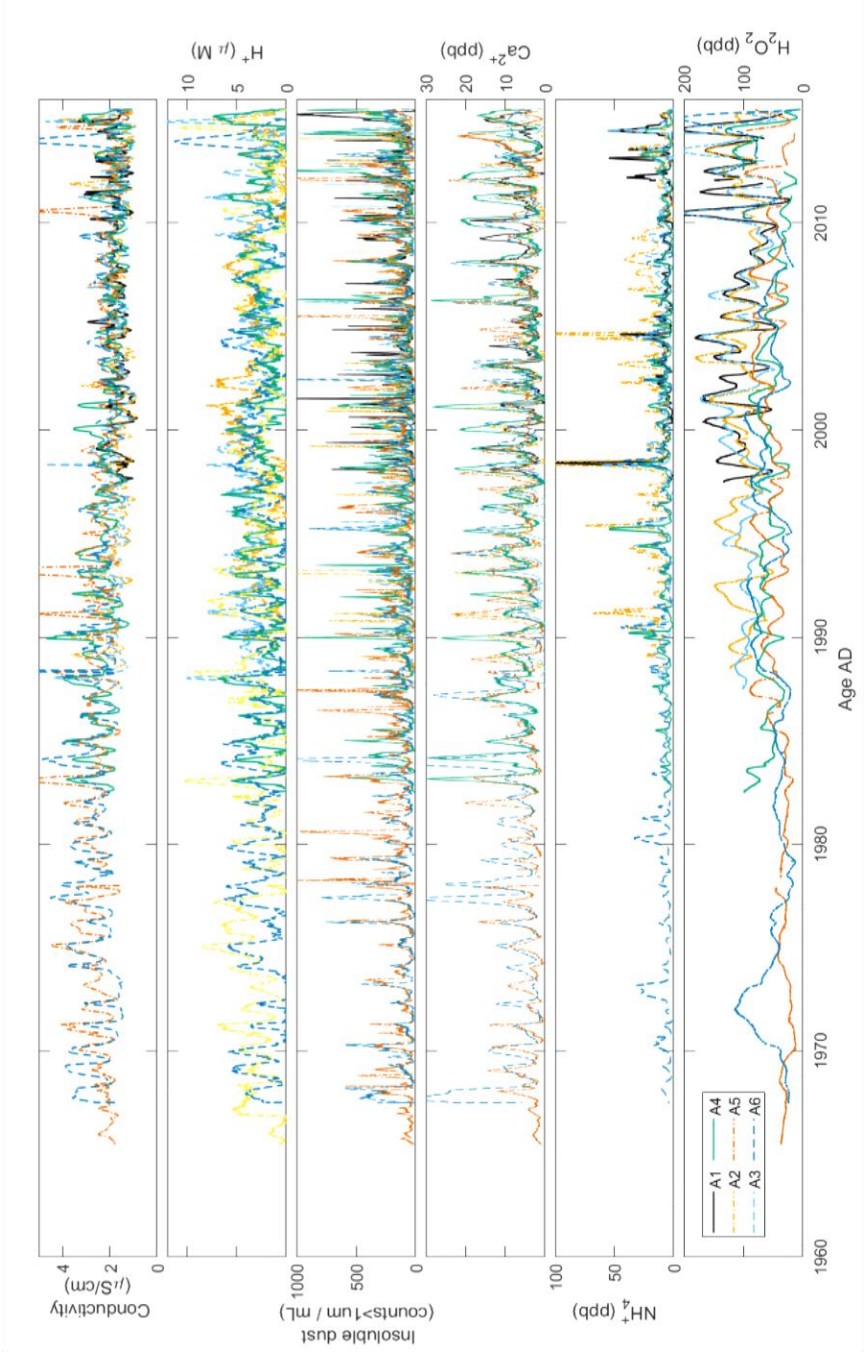

**Figure 2. Proxies measured by means of CFA in the six shallow firn cores; T2015-A1(black), T2015-A2(yellow dashed), T2015-A3 (light blue), T2015-A4 (green ), T2015-A5 (orange dashed) and T2015-A6 (dark blue dashed). From the top is shown conductivity, acidity, insoluble dust, $Ca^{2+}$, $NH_4^+$ and $H_2O_2$ concentrations. Note that for T2015-A1 the acidity and for T2015-A5 the $NH_4^+$ was not of sufficient quality and thus not shown.**

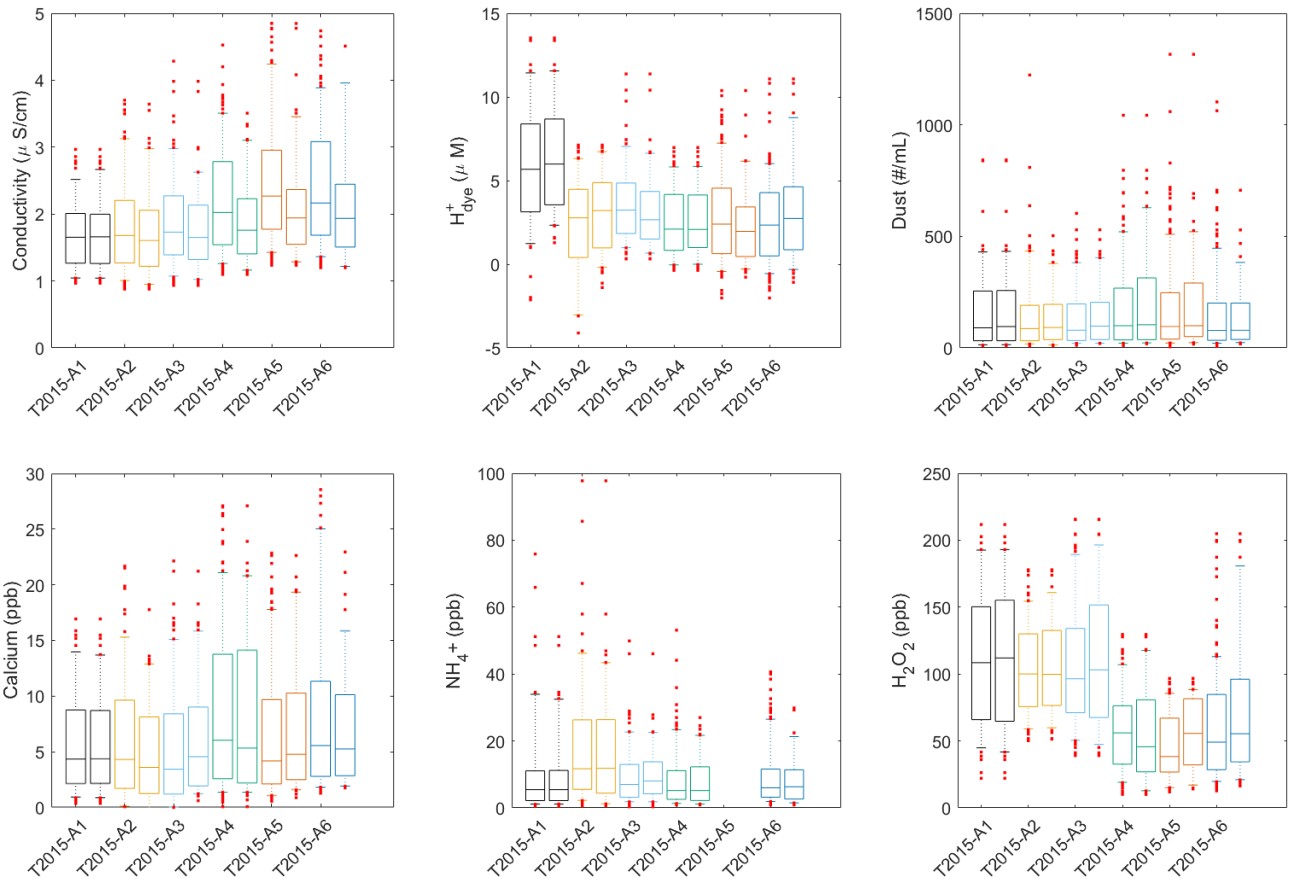

**Figure 3. Statistical representation of the monthly mean datasets. The central mark indicates the median, and the bottom and top edges of the box indicate the 25th and 75th percentiles, respectively. The whiskers extend to the 2.5 and 97.5th percentiles. In red dots are shown data exceeding the 2.5 and 97.5 percentiles. From left to right is shown T2015-A1(black), T2015-A2(yellow), T2015-A3 (light blue), T2015-A4 (green), T2015-A5 (orange) and T2015-A6 (dark blue). Two sets for each core is shown most left for the full temporal record available and right for the period 2000 AD and onwards to make comparable estimates between sites.**

increases in the $NH_4^+$ proxies. The $Ca^{2+}$ is comparable yet with a lower median than that found in the early Holocene for the NEEM deep ice core ( ~7ppb, Schüpbach et al. 2018). The inter-annual variations in the individual records are large for all proxies (whiskers in Figure 3).

Spatial concentration gradients (comparing 15-85%) in insoluble dust, $Ca_2^+$, $NH_4^+$, $H^+$, and conductivity are not easy to distinguish because of the inter-annual variability and the site specific noise. This despite the fact that the firn cores are spanning a distance of 426 km and accumulation is double or more in the northwest (T2015-A1, T2015-A2, T2015-A3, Table 1) compared to the central north and northeast (T2015-A4, T2015-A5, T2015-A6) (Kjær et al., 2021c). Gfeller et al. 2014 investigated several shallow cores at the NEEM site and reported that annual deposited aerosol concentrations in shallow firn cores can vary strongly over distances of a few meters. The study pointed out that one drill site could be representative for

>60% of the variability within a squared area of 100 m$^2$. We add that in Northern central Greenland for distances >100 kilometres apart significant median concentration changes between sites is not resolved beyond seasonal noise for insoluble dust, Ca$^{2+}$, nor NH$_4^+$. This suggest that the dust and NH$_4^+$ are mainly wet deposited in central Northern Greenland, producing similar concentration across all sites and suggest a single source area for each species far enough distant that individual weather events are not influencing the signal.

Contrary we observe a clear dependence on accumulation in H$_2$O$_2$ (Figure 3, Table S4, Table 1) with concentrations in the northwest (median 96-108 ppb) twice that in central and northeast Greenland (median 38-56 ppb) owing to the photolysis re-activation loss at the lower accumulation sites (Sigg and Neftel, 1988; Frey et al., 2006).

The conductivity also has spatial gradients and the median decreases from close to 2 µS cm$^{-1}$ (Figure 3, Table S4) in the low accumulation north-east (T2015-A4 and T2015-A5) to 1.60-1.66 µS cm$^{-1}$ at the higher accumulation sites west of the Greenland ice-divide (T2015-A1, T2015-A2, T2015-A3). We suggest this is an effect of the total dry deposited ions (*e.g.* sea salts) being more diluted in the west, but speculate that it could also originate from an anthropogenic (North American) input reaching first the western central ice sheet and only later when more dispersed the eastern. Unfortunately, the noise in the acidity records from the 6 firn cores is too large to help resolve if anthropogenic changes is the source and sodium were not analyzed for these cores.

Looking at the correlation between sites in the individual proxies (Supplementary section S3), we observe in general higher correlation values between the western cores (T2015-A1, T2015-A2 and T2015-A3), than the eastern or central cores probably owing to the better constrained dating at the high accumulation sites and in shorter records.

The monthly peroxide records correlate well for the high accumulation western sites (R$_{month}$ <0.56) as expected given that the dating is based on this proxy. Contrary, the eastern low accumulation sites H$_2$O$_2$ correlate less well as a result of the loss of signal. Also in the correlation of the annual H$_2$O$_2$ records this is the case, perhaps driven by similar overall accumulation pattern as speculated by *e.g.* Frey et al., 2006.

The Ca$^{2+}$ used to also constrain the dating for the low accumulation site also show significant positive correlations. However, T2015-A6 and T2015 -A5 are less well correlated to each other and to T2015-A1 and T2015-A2, respectively. Suggesting that the top part of T2015-A5 and T2015-A6 are perhaps offset in the precise assignment of summer months. The annual correlation for Ca$^{2+}$ and also insoluble dust is not significant, while insoluble dust monthly correlations is about half that found for Ca$^{2+}$. We interpret these lower correlations as either 1) individual dust depositions being more dispersed than similar Ca$^{2+}$, 2) that insoluble dust has an additional source on top of a shared source with Ca$^{2+}$ or 3) That the CFA analysis smoothing in calcium ensures better monthly correlations than that found for the better resolved insoluble dust, where individual deposition events thus can be recognized from each other.

The conductivity records are well correlated at either side of the ice divide in both the annual and monthly records; for the western cores R$_{month}$ >0.3, while in the east (including T2015-A6) R$_{month}$ >0.26. This supports the idea that the seasonal variation in conductivity could be driven by two different sources one in west and another one in the east. H$^+$ from ice cores are speculated to be the main controller of the conductivity in Greenland ice cores. Unfortunately, the quality of the H$^+_{dye}$ is not

sufficient to investigate if this is the case here. However, one could speculate in a North American $H^+$ source separate from a European influencing separately the western and eastern cores. We note also that sea salts were not analysed in this study, but that it in ice cores from coastal sites can be dominating the conductivity signal. Thus we speculate further, that the high monthly correlations in the conductivity records between eastern and western separately could be caused by sea salts from different open water sources, namely the Baffin Bay and the Greenland Sea respectively.

The ammonium records are generally well correlated between sites both for the annual ($R_{annual}$ 0.29-0.68) and monthly resolved records ($R_{month}$ 0.29-0.61), again more so at the western high accumulation sites ($R_{month}$ >0.38, $R_{annual}$ >0.42) suggesting a common source reaching all of northern Greenland.

Finally, we note that high resolution records, as in this study contain variations related not only to the climatology, but also to the analytical setup (eg. smoothing for the different CFA systems) and/or site specific noise and this noise limits the records capability to resolve spatial gradients between the firn records. Site specific noise is related to the local precipitation patterns, which can be disturbed by wind causing the formation of dunes, sastrugis or crust layers. These features mix up already deposited snow especially if precipitation is very event based. Melt layers at sites experiencing higher temperatures and ablation can also redistribute the deposited ions in the snow pack (Laepple et al., 2016; Gfeller et al., 2014).

## 4. Average seasonality

We remove the five year running average and in the excess investigate the seasonality by formal month (Figure 4). Thus the average seasonal cycle of excess concentration after removing the five-year mean contains also extreme events such as forest fires and volcanic horizons, which is discussed in more detail in section 5-Temporal trends. As $H_2O_2$ was used as the main cycle to date these cores we refrain from discussing its climatology. Still, we do note that for all sites except T2015-A6 the average seasonal cycle of $H_2O_2$ ( Figure 4, top left) is sinusoidal with the confidence intervals distributed almost equally above and below, where the maximum $H_2O_2$ concentration is defined as summer solstice (June), and the minimum is observed in the formal months December and January. However, for the low accumulation site (T2015-A6) it is evident that dating using just $H_2O_2$ was challenging and that the use of $Ca^{2+}$ have shifted the seasonal maximum value towards 2 months late (to ~August) and made the $H_2O_2$ seasonal cycle look less sinusoidal than seen at the other sites. This uncertainty on the dating is likely reflected in the average seasonality for other proxies in the T2015-A6 traverse core, and thus care should be taken interpreting especially T2015-A5 and T2015-A6.

### 4.1 Summer biosphere activity in Ammonium ($NH_4^+$)

$NH_4^+$ has a distinct maximum in the late spring and early summer months (April-June, Figure 4, top, second) catching the highest biological activity, while minimum concentrations occur in a wider part of the year from late autumn and early winter (Oct-Dec). The variability is high between the individual years (Figure 2 and Figure 3) and the annual maximum is wide and not very sinusoidal as evidenced in the seasonal cycle of the 15-85% quartiles (Figure 4, top, second). This is a result of an additional source in summer and early autumn namely the Canadian forest fires, and the uneven seasonal shape is evidenced

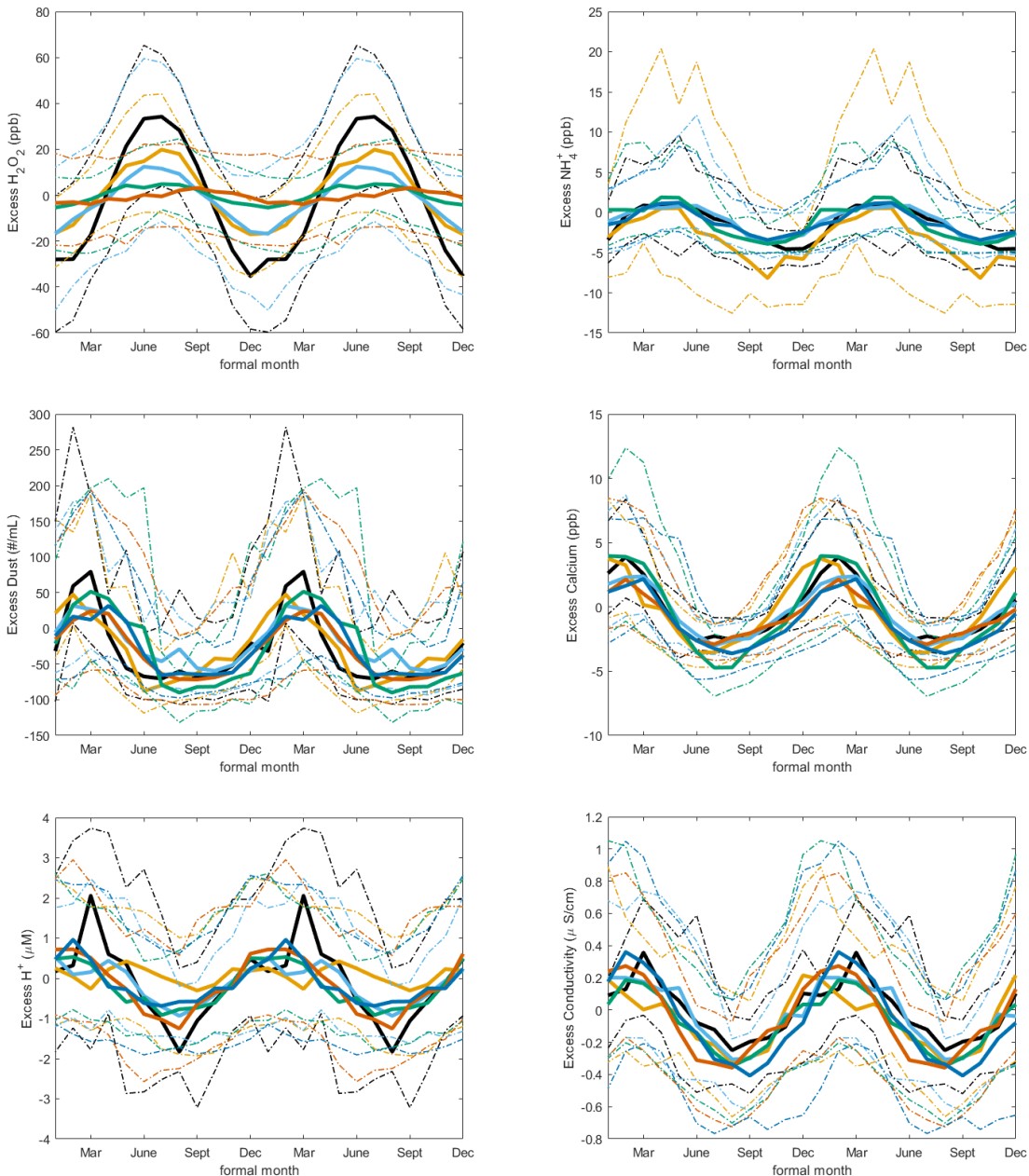

**Figure 4: From top left is shown H2O2, NH4+, Insoluble dust, Ca2+, H+ and Conductivity monthly medians (thick lines), based on the formal month definition, of the excess concentrations as defined after removing the 5 year temporal trend. Thinner dashed lines indicate the 15% to 85% quartiles of monthly averaged records. The colors reflect the six firn cores T2015-A2 (yellow), T2015-A3 (light blue), T2015-A4 (green), T2015-A5 (orange) and T2015-A6 (darker blue). Note that for T2015-A5 ammonium is not analysed.**

more so in the cores closest to the Canadian forest fire source area (T2015-A1 (Figure 4, black) and T2015-A2 (Figure 4, yellow)).

Gfeller et al. (2014) found median $NH_4^+$ to be largest in formal months June/July and that concentration maxima had shifted from preindustrial when it was as late as July/August for the NEEM site. We attribute the discrepancy to the uncertainty associated with the formal month definition and in differences in dating strategy, as we also note that the average $H_2O_2$ seasonal cycle observed by Gfeller et al. (2014) are shifted by 2 months compared to our assignment of annual maximum $H_2O_2$ in summer solstice.

## 4.2 Winter storms carry dust inland

Insoluble dust particles and $Ca^{2+}$ as its soluble compound (Figure 4, middle panels), coming from e.g. $CaCO_3$, or $CaSO_4$, are common paleo-climatological proxies for global aridity and wind strength. The average $Ca^{2+}$ seasonal cycles (Figure 4, middle right) in the traverse cores show a late winter/early spring maxima ($Ca^{2+}$ Jan-March, insoluble dust Feb-May) as also observed by others (Kang et al., 2015; Kuramoto et al., 2011; Amino et al., 2020). Minimum concentrations are found in the summer months July and August. The $Ca^{2+}$ seasonal cycle is smooth compared to that of the insoluble dust, where we observe high insoluble dust loads also in the adjacent months of the annual maximum as evidenced by the monthly 85% quantile (Figure 4, middle left). In the cores T2015-A4, T2015-A5 (EastGRIP site) and T2015-A6 (central divide) it looks like insoluble dust is deposited twice a year (early spring and late autumn/early winter). Whilst this may be due to a local source as was speculated in other areas of Greenland (Amino et al., 2020; Bullard and Mockford, 2018; Nagatsuka et al., 2021), it could also be ascribed the fact that deposition events are rare in north Central Greenland (McIlhattan et al., 2020) and thus the dust maxima could be found in other formal months.

## 4.3 Spring Arctic haze in acidity and conductivity

Maximum concentrations of acidity (Figure 4, bottom left) are recorded in early spring (March), however with a wide distribution in the adjacent months (January-May). This is in line with previous findings where the seasonal maxima of acidity is found in spring and attributed the modern Arctic haze phenomenon, with anthropogenic pollutants such as $SO_2$ building up in the atmosphere during stable and dry winter conditions being deposited in spring, when precipitation rates grow (Gfeller et al., 2014; Kuramoto et al., 2011; Quinn et al., 2007). As the conductivity is mainly driven by the $H^+$ (Kjær et al., 2016) its annual maxima concentration is in close proximity (Figure 4, bottom right), however one month shifted towards an earlier deposition due to the influence likely from sea salts. Similar observations of seasonality have been made at the NEEM site (Jan-Apr); and Humboldt North sites (Dec-March) (Gfeller et al., 2014; Pasteris et al., 2012).

## 5 Temporal trends

In the following we investigate temporal trends observed in the six records. We find a clear signal of the 1970's acid contamination in the conductivity (Figure 2) and increases in the intermediate and large insoluble dust fluxes (Figure 5) suggesting an activated transport of local Greenland dust to the Northern Greenland Ice Sheet, more so at lower altitude sites closer to the coast.

### 5.1 Anthropogenic increase in the 1970's and 1980's observed in the conductivity

The 1970's increase in conductivity associated with anthropogenic sulphur and $NO_x$ emissions has previously been observed in firn records from Greenland (Fischer et al., 1998; Kjær et al., 2016; Pasteris et al., 2012). For our oldest records, T2015-A5 and T2015-A6, a decrease in the conductivity between the early part of the record (1960's-1990s) and the younger part confirms the effect of mitigation measures (Figure 2 and supplementary Figure S8 top). The conductivity in polar ice cores is found to be mostly controlled by $H^+$ (Kjær et al., 2016). We would thus expect similar trends of anthropogenic contamination in the acidity record. Unfortunately, the inter-annual variability in the acidity record is large making it difficult to assess the temporal trend (Figure 2, and supplementary Figure S8, bottom and Table S4). This is mainly a result of the measurement technique being subject to flow sensitivity (Kjær et al., 2016), but also influenced by individual concentration maximas associated with volcanic events (discussed in section 6.1) .

### 5.2 Local dust activation

Recent publications have suggested that with warming and current mass loss in Greenland, local sources of dust are activated. For the Holocene large sized particles of local dust are observed in coastal ice core sites (Marius Folden Simonsen et al., 2019) and at the SE-Dome core (South east Greenland) an increase in the local sourced dust flux in the period 2000-2010 compared to 1960-2010 was evidendent as an increase in the larger particles (>5µm) in the autumn (Amino et al., 2020). In the west (Kangerlussuaq) local dust activation in the period 2000-2010 have been observed (Bullard and Mockford 2018) and in the North-West ice core Sigma-D the period 1915–1949 and 2005–2013 had a mineral composition suggesting a west Greenland source, while the remainder of the past 100 yrs suggested a Canadian dust source area suggested related to warmer temperatures in Greenland activating the local dust source areas (Nagatsuka et al., 2021).

When comparing the period between 2000 and 2015 to that of the full firn cores we do not observe any significant increase in the total number of insoluble particles at any of the 6 northern sites studied nor in the $Ca^{2+}$ concentrations (Figure 3, Table S4). The annual insoluble dust flux (Table 2 and Figure 5) was determined assuming all spheres are perfectly round, using a mean density of 2400 kg/m3 of the dust and annual accumulation for the 6 sites (Kjær et al., 2021c). We note that uncertainty associated with the flux calculation is related to the accumulation and uncertainties within the timescale, as well as the assumption of spherical dust and that the following discussion must not be over-interpreted. We find a total dust flux between 0.016 and 0.057 mg kg$^{-1}$ yr$^{-1}$ (Table 2) with the smaller fluxes at the central higher altitude sites (T2015-A3 and T2015-A6) as

anticipated. We further split the data into three bins; small (1.25 to 2.9 μm radii), intermediate (2.9 to 8.13 μm) and large (8.13 to 10.5 μm) (Simonsen et al., 2019). The largest particles (>10.5 μm) are omitted from further analysis as they are subject to poor statistics and the smallest sizes (<1.25 μm) as well as they are noisy. We find that by parting the dust data this way we have 12-29% of the total dust in the small range, 44-52% in the intermediate range and in the large range just 3-13% of the

5   total insoluble dust flux observed in the cores over the period 1998-2015. For the central cores, T2015-A3 and T2015-A6, the large particle fluxes are just 3 to 6% of the total suggesting that the large particles do not make it to the high central ice cap to the extent that it does the lower altitude sites.

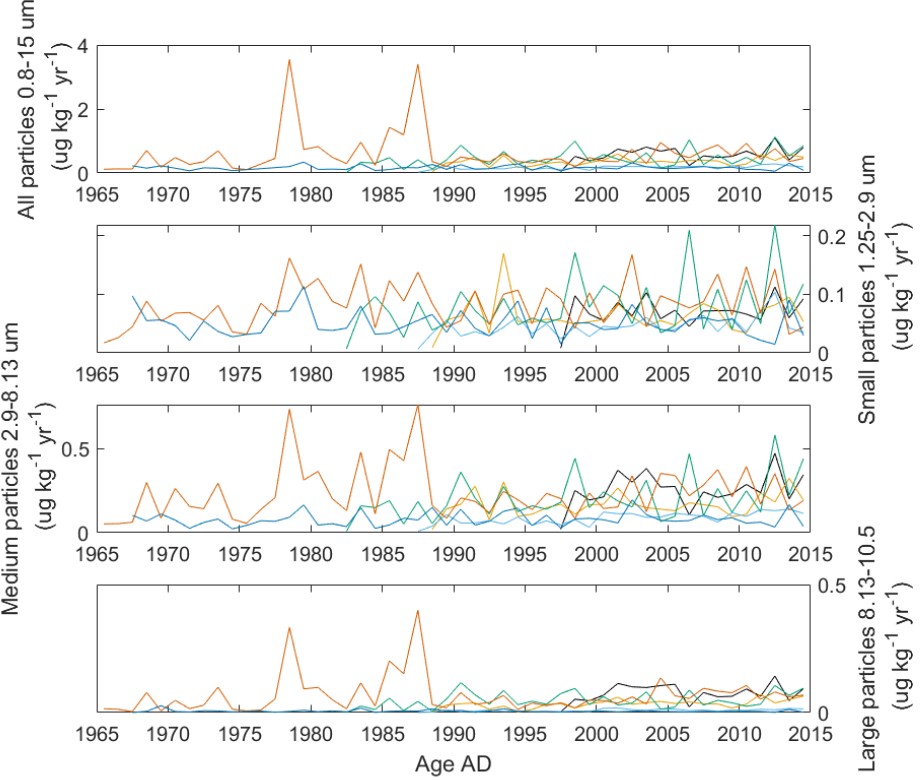

**Figure 5: Insoluble dust particle fluxes determined from the Abakus instrument by assumptions of perfect spherical particles and a**
10  **weight of 2400 kg/m³. From the top total 0.8-15 um, Small particles 1.25-2.9 um, Medium particles 2.9-8.13 um and Large particles 8.13-10.5. The colors reflect the firn cores T2015-A2 (yellow), T2015-A3 (light blue), T2015-A4 (green), T2015-A5 (orange) and T2015-A6 (darker blue).**

In the following we investigate the trend in the dust flux over the period 1998-2015 (Table 2 - right hand side, Figure 5, Figure S9). First of all, we note that the record is noisy and that the p-values (Table 2) in what we discuss next are high (significant

15  $p<0.2$). In all but the central core T2015-A6, we observe an increase in the total insoluble dust flux (ug kg$^{-1}$ w.eq) of 1.1 % to 3.0 % a year (Table S7) since 1998, significant ($p<0.2$) for the cores T2015-A1, T2015-A2, and T2015-A3. Excluding T2015-A6 and T2015-A5, we note that the small particle fluxes increase between 0.9 % and 3.0 % annually, significant for T2015-A3 ($p<0.2$). Intermediate dust sizes increase every year by between 1.7 % and 3.4%, significant only for T2015-A3 and the

large bins increase by 0.98 % to 3.96 % annually for T2015-A2 and T2015-A5. This suggests that the increasing trend (in percent) is larger at the intermediate and large particle sizes, which we interpret to reflect an increased activation of local dust in northern Greenland over the period 1998-2015. While we once again emphasize that these results are very influenced by accumulation and thus the dating of the cores, we interpret these changes as an activation of local sources in Greenland or

Northern Canada supporting the results found by others.

**Table 2: Dust fluxes and trends in dust fluxes for the period 1998-2015 for each of the 6 shallow firn cores. Small (1.25-2.9 μm), intermediate (2.9-8.13 μm) and large (8.13-10.5 μm) refers to the dust sizes as analyzed by the Abakus instrument. P values given in parenthesis, when significant p<0.2 in bold and when p<0.1 in bold and italic.**

| Firn core | Dust Flux [μg/kg/yr] | | | | Dust flux trend 1998-2015   [μg/kg/yr2] (p-value) | | | |
|---|---|---|---|---|---|---|---|---|
| | Total | Small | Intermed | Large | Total | Small | Intermed | Large |
| **T2015-A1** | 57.46 | 7.00 | 25.60 | 7.55 | **1.74 (0.11)** | 0.13 (0.25) | 0.64 (0.17) | 0.19 (0.24) |
| **T2015-A2** | 38.71 | 6.16 | 17.28 | 4.01 | **0.72 (0.15)** | 0.09 (0.27) | 0.29 (0.27) | **0.10 (0.11)** |
| **T2015-A3** | 19.24 | 4.84 | 10.06 | 1.04 | *0.53 (0.01)* | *0.15 (0.08)* | *0.34 (0.00)* | 0.03 (0.21) |
| **T2015-A4** | 53.78 | 9.66 | 24.46 | 5.39 | 0.61 (0.65) | 0.09 (0.74) | 0.62 (0.74) | 0.05 (0.70) |
| **T2015-A5** | 55.30 | 8.69 | 23.91 | 6.45 | 1.32 (0.21) | 0.00 (0.99) | 0.47 (0.31) | *0.26 (0.04)* |
| **T2015-A6** | 16.02 | 4.58 | 8.18 | 0.43 | -0.09 (0.79) | -0.00 (0.98) | -0.02 (0.91) | 0.00 (0.84) |

# 6 Extreme events

In the following we investigate the extreme events (>0.97 quantile) observed in conductivity, acidity and $NH_4^+$ after removing a five year running median.

## 6.1 Spatial distribution of recent Volcanic events- excess acidity and conductivity.

Local acid or conductivity concentration maxima exceeding the background in ice cores are often interpreted as volcanic eruptions and used for constraining ice core ages (Sigl et al., 2016b; Svensson et al., 2008; Vallelonga et al., 2014; Kjær et al.,

2016). In addition, such acid markers can be used to determine volcanic climate forcing from volcanic eruptions used in climate modelling (Gao, Robock, and Ammann 2008; Robock and Free 1995). In Figure 6 extremes in the acidity/conductivity determined by concentrations exceeding 97.5% quantiles from the 5 yr running average based on monthly means of each individual record are shown and interpreted as volcanic horizons. Vertical bars in green, yellow and turquoise show which eruptions were identified in conductivity, acidity or both proxies respectively for each of the 6 firn cores, thus providing an

overview of the spatial distribution in northern Greenland of specific acid plumes.

Already from a first look it is evident that not all plumes distribute to the entire northern Greenland and thus we confirm the previous findings (Robock and Free 1995; Gao, Robock, and Ammann 2008) that a single core can't be used if aiming to create a record of volcanism from ice cores. Further using the conductivity alone can be deceiving if aiming to find volcanic horizons. We also emphasize our dating was restricted to use mainly $H_2O_2$ and $Ca^{2+}$, and thus we have not made any volcanic

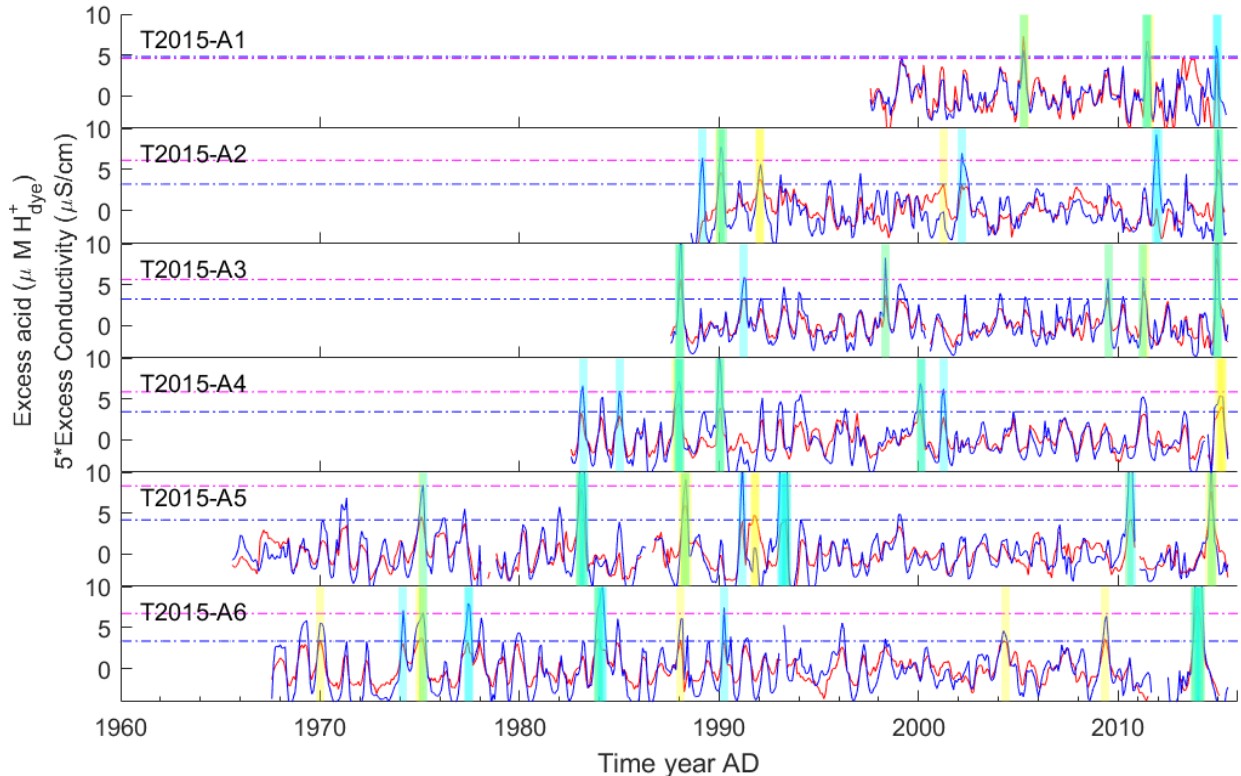

**Figure 6: Excess acidity (red) and Excess conductivity (blue) as compared to a 5 year running average. Note that the conductivity is scaled with a factor 5. Dashed horizontal lines indicate 97.5% quantile for conductivity (light blue) and acidity (yellow), both (green). Vertical bars indicate times exceeding 2 standard deviations for conductivity and acid compared to a 5 year running average in yellow and cyan respectively, green when observed in both. From top is shown T2015-A1 through to T2015-A6 in the bottom.**

matching between the records as part of the dating. Also we note that other markers are more specific to volcanic eruptions than the ones used in this study, *e.g.* non sea salt sulphate or S isotopes (Severi et al., 2012; Sigl et al., 2016a; Mayewski et al., 1990; Lin et al., 2022; Crick et al., 2021).

The eruptions that are observed in more than one of our traverse cores are presented in Table 4; Below we discuss the sources for the acidity and conductivity reference horizons observed in the firn cores in more detail.

**2014/2015-Bardabunga-Holuhraun, Iceland.** In late August 2014 until February 2015 the Bardarbunga fissure, also known as the Holuhraun eruption (Volcanic eruption index –VEI 0) took place in Iceland. The SO2 emissions are estimated to have been $10.7 \pm 3.0$ Mt (S.R. Gíslason et al., 2015). We observe elevated acidity in all of our firn cores of between 4.3 to 12.1 uM $H^+$, but note that T2015-A6 is offset compared to the other records due to uncertainty in the depth registration of the top.

T2015-A3, T2015-A6 and T2015-A5 show the highest acidity concentrations from the Holuhraun eruption while T2015-A2, T2015-A4 and T2015-A1 are less pronounced. At the EastGRIP site previous results from snow pits also evidence the eruption (Du et al., 2019a). The fact that this eruption is found at all 6 sites and also in various other north Greenland sites (Kjær et al., 2021a) suggest that this eruption horizon is deposited in the wider Central Northern Greenland. Despite the modelled $SO_2$ and $SO_4$ plume trajectories from the eruption by e.g. (Boichu et al., 2019) not reaching Greenland.

**Table 4: Volcanic eruptions. Suggestion for volcanic sources of the years where excess acidity or excess conductivity as compared to a 5 year running average exceed 97.5% quantile in each of the six shallow cores. In addition is shown other shallow cores with similar excess acid, conductivity or sulphate. Dashed line means data is not available. Years in parenthesis indicate that while there is an increase compared to background it does not exceed the 97.5% quantile. 1-Pasteris et al., 2012, 2-Zielinski et al., 1994, 3-Fischer et al., 1998, 4-Sigl et al., 2016a; Kjær et al., 2021b, 5- Du et al., 2019b; Kjær et al., 2016.**

| Source | T2015-A1 | T2015-A2 | T2015-A3 | T2015-A4 | T2015-A5 | T2015-A6 | Humboldt North[1] | GISP2[2] | NGT27[3] | NEEM site[4] | *EGRIP site[5]* |
|---|---|---|---|---|---|---|---|---|---|---|---|
| 2015 Bardabunga (VEI 0) | 2015 | 2015 | 2015 | 2015 | 2015 | (2014) | - |  | - | 2015 | 2015 |
| 2011 Grimsvotn (VEI 4) | 2011 | 2011 | 2011 | (2011) | 2010 |  | - | - | - | - | - |
| 1998 Grimsvotn (VEI 2) | (1999) |  | 1998 |  |  |  | - |  | - | - | - |
| 1991 Pinatubo (VEI 6) or 1991 Hekla (Iceland) | - | 1991 | 1991 |  | 1991, 1992 |  | 1991 |  | 1992 | 1991,1992 | No |
| 1989 Redoubt (VEI 3) | - | 1989 |  | 1989 |  | 1990 | No |  | 1989 |  | 1989 |
| 1987 *Cleveland (VEI 3) or Kamchatka (VEI4)* | - | - | 1987 | 1987 | 1987 | 1987 | 1987 |  |  |  | 1986 |
| *1983 Grimsvotn, or 1982* Mexican El Chicon (VEI5) | - | - | - | 1983 | 1983 |  | 1983 |  |  |  | 1982 |
| 1977 Krafla fires | - | - | - | - | (1977) | 1977 | 1977 | 1978 | 1977 |  | No |
| 1974 Russian Kliuchevskoi-Kamchatka (VEI 3) | - | - | - | - | 1974 | 1974 | No |  |  |  | No |
| 1971 Hekla (or forest fire event) | - | - | - | - | (1971) | 1970 | 1971, 1972 | 1971 |  |  | 1971 |

**Grimsvötn 2011 (VEI4), Iceland.** Another concentration maxima (6.7-8.6 uM) observed through most the firn cores, except T2015-A6 central Greenland core, is the Grimsvötn 2011 (VEI4), Iceland. The eruption took place from 21-28[th] of May 2011(Hreinsdóttir et al., 2014) and had major societal impact as 900 flights were cancelled in Europe, despite the fact that the plume of particles stayed mostly local and turned northward. Modelling suggests that the particles travelled mostly westward reaching as far as Finland, while the sulphuric acid moved at a higher level northward toward Greenland, where it turned westward crossing over Greenland. The traverse records evidence that acid from the Grimsvötn eruption was deposited even further north than found in the modelled plumes (Moxnes et al., 2014; Kerminen et al., 2011; Petersen et al., 2011).

**Eyjafjallajökull (VEI4), Iceland.** The famous Eyjafjallajökull eruption (VEI4) between 14 April–23 May 2010 is not strong enough to enhance the acidity above background variability, except for T2015-A5. This is expected as the plume from this

eruption had a south-eastward track down over Europe (Schumann et al., 2011; Thomas and Prata, 2011), and only after a major turnaround towards the northern globe the acid reached the North west Greenland.

**Grimsvötn 1998 (VEI2), Iceland.** The 1998 Grimsvotn event extended for 10 days long in December with a plume up to 10 km altitude. An increase in is observed in the firn T2015-A2 in late 1998 and early 1999, suggesting that the plume from the eruption made it to the northern central Greenland.

**Pinatubo 1991 (VEI6), Philippines.** The June Pinatubo has previously been reported both at the Greenland NEEM site and the Antarctic WAIS site (Sigl et al., 2016b) is only observed in the T2015-A2, T2015-A3 and T2015-A5 and thus its use as a synchronisation event between hemispheres might be limited as it does not show in all our Greenland cores. We also note that Pinatubo was not significantly found in the NEGIS record (Vallelonga et al., 2014; Kjær et al., 2016) close to our T2015-A5 and T2015-A4 records. Further, we recall that a small eruption from the Icelandic Hekla took place in 1991 (0.02 $km^3$ tephra (Thordarson and Larsen, 2007)) and suggest that as an alternative source of what is observed in Greenland cores in 1991.

**Redoubt 1989 (VEI3), Alaska** The Redoubt eruption 1989 (VEI3) in southern central Alaska consisted of more than 20 individual eruption events that began in December 14$^{th}$ 1989 and lasted until late 1990 (Casadevall, 1994; Scott and McGimsey, 1994). During the eruption sulfur dioxide emission rates were between 800-6600 metric tons a day, with the highest emissions in March and tephra plumes are documented to altitudes of 7-10 km (Brantley, 1990; Scott and McGimsey, 1994). We observe an imprint in T2015-A2, T2015-A4 and T2015-A6 of what could be the Redoubt eruption, with concentrations of 8.7-10.3 μM $H^+$.

**Cleveland 1987 (VEI3), Alaska.** The Aleutian Iceland eruption took place in late 1987 and lasted for about 2 months and a signature is distributed through all our records in 1987. However, we note that Cleveland is an active volcano with at least 22 eruptions over the past 230 years and other VEI 3 eruptions from Cleveland happened in 1994, 2001, 2006, with multiple smaller eruptions in between. We note that in T2015-A2 an event exceeding 97.5% is observed also in early 2002. We wonder about the lack of signal in the other firn cores from these large eruptions from Cleveland and speculate that the source in 1987 may not be Cleveland. Other sources of large volcanic eruptions in the period 1986-1987 include Mount Augustine 1986, Alaska (VEI 4), which also erupted in 2005/2006 (VEI3) and the Russian eruptions of Chikurachki-Kuril Islands and Kliuchevskoi-Kamchatka in 1986 (VEI 4) and 1986 (VEI 4) respectively.

**Grimsvötn 1983, Iceland** In May 1983 a plume of 10 kt $SO_2$ rose up to 8 km from the Grimsvötn volcano, and in all cores covering that period we observe a significant excess acidity for that year, again suggesting a central Greenland route. Yet, this eruption only lasted 2 days. Another explanation for an excess 1983 concentration in Greenland ice sometimes invoked is the spring VEI 5 Mexican El Chicon 1982 eruption (Palais et al., 1992).

When looking further back in our records, we note that one has to be careful when assigning excess acidity to volcanic events as the period 1970-1980's is highly influenced by anthropogenic sulphates. Thus even after removing the mean background signal, spring Arctic haze events can be very high in sulphates and resemble volcanic strata. Regardless, we note the acidity and conductivity maxima's that are found between the T2015-A5 and T2015-A6 sites reaching furthest back in time, and compare with the nearby NEGIS (Kjær et al., 2016; Vallelonga et al., 2014) and Humboldt North records (Pasteris et al., 2012).

**1977 Krafla Fires** The so-called Krafla Fires which consists of 9 individual eruptions took place in Iceland between 1975 and 1984 (Thordarson and Larsen, 2007). We do not clearly observe the Krafla Fires in the records of T2015-A5. However, T2015-A6 show excess in late 1977 and we note that four rifting events occurred in the period October 1976 as part of the Krafla fires, with the last one in September 1977, which included an explosive event through a geothermal borehole producing 26m3 of tephra (Thordarson and Larsen, 2007). In 1980's similar rifting occurred, but they are not mirrored in our Greenland records. We also note that (Kjær et al., 2016) observed an increase in the conductivity for the NEGIS core in 1977 and similarly was observed at the Humboldt North site (Pasteris et al., 2012), suggesting that the plume took a route over central north-east Greenland. The last eruption of the Krafla fires was September 1984 and it may have contributed to the peak assigned to Grimsvötn above.

**1974 Kliuchevskoi-Kamchatka (VEI3), Russia.** We observe excess acidity in 1974 for both T2015-A5 and T2015-A6 sites confirmed also at the Humboldt North site (Pasteris et al., 2012). We suggest it could be the VEI 3 from the Russian Kliuchevskoi-Kamchatka.

**1970 Hekla, Iceland.** In T2015-A6 1970 we find what potentially is the Hekla 1970 eruption (0.07 tephra km$^3$ 0.03 tephra DRE km$^3$ (Thordarson and Larsen, 2007)). A similar event is not reflected in T2015-A5 although the event is not large enough to exceed the 97.5% boundary. However, both in the NEGIS core close by T2015-A5 and in the Humboldt North core an increase in conductivity and acidity is observed in 1970's (Pasteris et al., 2012; Kjær et al., 2016; Vallelonga et al., 2014).

In conclusion we find several recent horizons that can be attributed mainly to Icelandic volcanoes, but we note that many years have a corresponding large (>VEI3) eruption in the regions of the Bering Sea region (Russian and Canadian Arctic) and thus the direct assigning of specific volcanoes is not straightforward when based solely on the excess acidity or conductivity. Some of the significant acidity and conductivity horizons observed across the 6 traverse cores are also seen in Humboldt North and NEGIS ice cores (Kjær et al., 2016; Pasteris et al., 2012; Vallelonga et al., 2014). We find that the modelled plumes for recent eruptions often do not reach the sites in which we find the acidity deposited and suggest that such dispersion model long term transport could be improved.

## 6.2 Canadian burned land area observed in extreme NH$_4^+$ events

The number and extent of fires through time of both natural and anthropogenic origin have varied and several fire index records exist. Extremes in NH$_4^+$ in ice cores have been used as a proxy for North American forest fires (Legrand et al., 1992; Zennaro et al., 2014; Fuhrer et al., 1996) while the background NH$_4^+$ is related to biogenic emissions from soil and vegetation and thus temperature on a longer timescale. Other commonly used fire proxies in ice cores include formate, which is found well correlated with excess NH$_4^+$ (Legrand et al., 1995; Savarino and Legrand, 1998), levoglucosan, which is specific to biomass burning events, black carbon, which is also subject to other anthropogenic sources (Zennaro et al., 2014; Segato et al., 2021), dehydroabietic acid (Parvin et al., 2019) and vanillic acid (Grieman et al., 2018a; Kawamura et al., 2012; Grieman et al., 2018b). The amount of fires as determined in ice cores have been found to vary over time with an increase in the mid-1600s (Zennaro et al., 2014) and it is speculated that current climate change and anthropogenic activity could enhance fires.

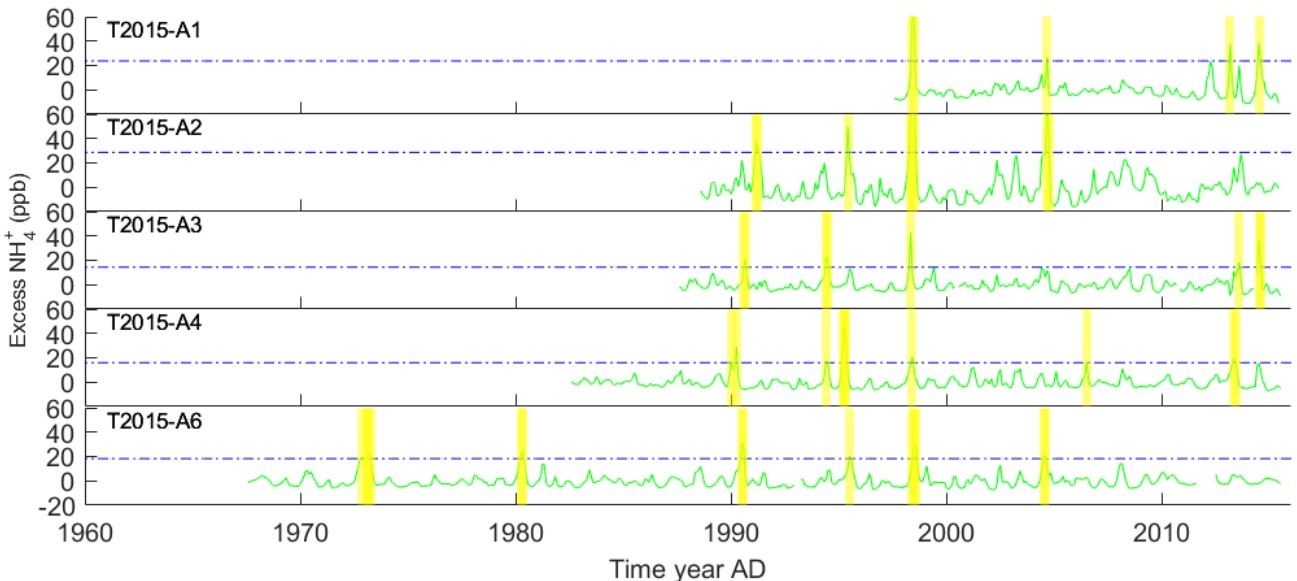

Figure 7: Excess NH$_4^+$ (green) as compared to a 5 year running average. Vertical bars (yellow) indicate times exceeding the 97.5% quantile (horizontal blue dashed). From top is shown T2015-A1 through to T2015-A6 in the bottom, note T2015-A5 was not analysed for NH$_4^+$.

Here we assess excess NH$_4^+$ (exceeding 97.5%) as a proxy for forest fires in the 5 traverse cores and compare with fire records from other recent ice cores (Parvin et al., 2019; Zennaro et al., 2014; Gfeller et al., 2014; Legrand et al., 2016; Pokhrel et al., 2020). We use NH$_4^+$ excess as a proxy for forest fires, however, bear in mind that NH$_4^+$ extremes was found to only replicate 8 out of 14 levoglucosan peaks at the NEEM site (Zennaro et al., 2014). Furthermore, for the NEEM site, Gfeller et al. 2014 observed that as a result of wind reworking, a single core close by 5 other cores only captures 70 to 80 % of the interannual

variability of the reconstructed NH$_4^+$ atmospheric aerosol load.

We start out by noting that the excess NH$_4^+$ (after removing the 5 year running average, Figure 7) from the individual traverse records correlate well (Pearson correlation R>0.4, p>0.01, Table S5) to each other in the central and west. West of the ice divide (T2015-A1, T2015-A2 and T2015-A3) annual correlations are as high as R=0.82 (between T2015-A1 and T2015-A3), while correlations between the western core T2015-A4 and the T2015-A6 central core is lower. We continue to make a

combined fire proxy record for the 5 cores by normalizing for each of the records the annual excess value and taking the mean of the cores covering the years. We observe that the combined fire proxy record (Figure 8) correlates with the Canadian National Forestry Database records of Forest burned area (Parisien et al., 2012; Canadian Forest Service, 2013) by R=0.49 (p=3.74·10$^{-4}$) from 1987 onwards and for the period 1959-2015 R=0.48 (p=0.009). This suggests that the combined traverse record of excess NH$_4^+$ is a fair proxy of burned area in Canada (Figure 8). Omitting T2015-A4, the only eastern core we find

that the central and western correlate with the Canadian forest fire record by 0.45 (p=0.001, 1987 onwards), suggesting that adding even the eastern records improve the fire proxy.

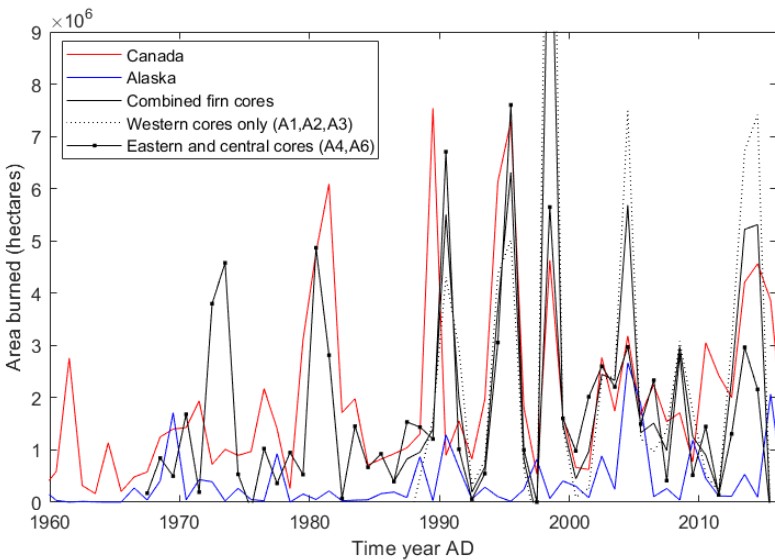

**Figure 8 Forest fire composite (black full line all records, dashed only western cores, with dots only eastern cores) compared to in red Canadian (Parisien et al., 2012; Canadian Forest Service, 2013) and in blue Alaskan forest fire indexes (AICC - Predictive Services - Intelligence / Reports, 2021; Legrand et al., 2016).**

Below we discuss the individual years of high $NH_4^+$ concentration in comparison with other ice core records starting from the oldest extreme $NH_4^+$ layers and speculate on sources based on the Alaskan (AICC - Predictive Services - Intelligence / Reports, 2021) and Canadian fire indexes (Canadian Forest Service, 2013). In Table 5 the years that contain extreme $NH_4^+$ is presented in the 6 firn cores investigated here as well as other fire tracers from other ice cores together with the suggested source. (>97.5% of each full record).

**1972 Russia** The 1973 extreme $NH_4^+$ (T2015-A6) was also observed in the Kamchatka ice core (1972) and NEEM ice cores (1973), but not in southern D4 nor the central Summit cores (Zennaro et al., 2014; Kawamura et al., 2012; Legrand et al., 2016) and it is speculated to originate from Russian fires resulting from droughts in 1972.

**1980** (>4.5 Mha) **and 1981 (>6 Mha) Canada** The 1980-1981 events have previously been observed in the $NH_4^+$ record around NEEM (S1 and main core) and in the Northern Greenland Tunu vanillic acid record (Grieman et al., 2018b; Gfeller et

al., 2014). The signal is also observed in our core T2015-A6 close to the ice divide, showing that the signal of the Canadian fires in 1980 (>4.5 Mha) and 1981 (>6 Mha) is widespread in Northern Greenland, and not wider Greenland as it is neither observed in the D4 nor the central Summit cores (Legrand et al., 2016).

**1990's three events Canada (>4.5 Mha)** The pattern of three extreme $NH_4^+$ events between 1989 and 2000's is observed in all records covering the period (T2015-A2, T2015-A3, T2015-A4, T2015-A6). The same pattern was observed in the South

Eastern SE-Dome core (Dehydroabietic acid, and low levoglucosan) and at the NEEM site provided the dating uncertainty of +/-1 year is taken into account (Gfeller et al., 2014). In addition the extreme in 1994 was also observed at Summit station with an increase in $NH_4^+$ and formate (Legrand et al., 2016) making the 1994 signal Greenland wide (Parvin et al., 2019), while the

**Table 5. Forest fires. The years where high excess ammonium have been observed exceeding the 97.5% quantile in the six firn cores is noted, together with observations from additional firn and snow cores and suggested sources. Dashes indicate that data is not available at the year in question, while "No" indicate that a fire is not seen in that year in a record. 1-Levoglucosan (Pokhrel et al., 2020), 2- (Kawamura et al., 2012), 3- NH4+, levoglucosan, Black carbon (Gfeller et al., 2014; Zennaro et al., 2014), 4-levoglucosan or vanillic acid (Grieman et al., 2018b), 5- NH4+(Legrand et al., 2016), 6-Levoglucosan, dehydroabietic acid (Parvin et al., 2019).**

| Source | T2015-A1 | T2015-A2 | T2015-A3 | T2015-A4 | T2015-A6 | Aurora peak, Alaska[1] | Ushkovsky, Kamchatka[2] | NEEM[3] | TUNU[4] | D4[5] | Summit[5] | SE-Dome core[6] |
|---|---|---|---|---|---|---|---|---|---|---|---|---|
| *2012-2014 Canada (>3.5 Mha)* | *2013, 2014* | *No* | *2013, 2014* | *2013* | *No* | - | - | - | *2012-2014* | - | - | *2013-2014* |
| 2004 Canada (~3 Mha), or 2004 Alaska (2.4 Mha) fire, | *2004* | *2004* | *No* | *No* | *2004* | *2005* | - | *2003, 2004, 2005* | *2004-2007* | - | - | *2003* |
| 1998 Canada (4.5 Mha) | *1998* | *1997* | *1998* | *1997* | *1998* | *1999* | - | *1999* | *1996-1998* | - | - | *1998* |
| 1994, 1995 Canada (>6 Mha) | - | *1995* | *1994* | *1995,1994* | *1995* | *No* | - | *1996* | *No* | - | *1993-1994* | *1995-1996* |
| 1989 Canada (>7.5 Mha) | - | *1991* | *1990* | *1990* | *1990* | *No* | *1989* | *1991* | *No* | *No* | *No* | *1988-1989* |
| 1980 Canada (>4.5 Mha) and 1981 (>6 Mha) | - | - | - | - | *1980* | *No* | *1981* | *1980* | *1980* | *1980* | *1980* | *1981* |
| 1972 Russia | - | - | - | - | *1972/1973* | *No* | *1972* | *1973* | *1972* | *No* | *No* | *1973* |

1990 NH$_4^+$ layer is not. In the Canadian burned area record 1989 (>7.5Mha), both 1994 and 1995 (>6Mha) and 1998 (4.5Mha) are significant, making Canadian fires the likely source areas for the events observed widely in Greenland in the 1990's.

**2004 Canada (~3 Mha)** 2004 is less significant in the Canadian burned area record (~3 Mha), but we note that the Alaskan Taylor Complex (2.4 Mha) fire, which is the largest in Alaskan records since 1940, might also add to the signal observed in central north Greenland (T2015-A1, T2015-A2 and T2015-A6). In 2005 elevated levoglucosan was also found in an Alaskan ice core (Pokhrel et al., 2020) and NH$_4^+$ have previously been observed in firn cores from NEEM (Gfeller et al., 2014), and in the SE-dome ice core dehydroabietic acid concentration is elevated above background, suggesting that the 2005 fire deposition is widely spread over Greenland, despite not being observed in all the firn cores presented here.

**2012, 2013, 2014 Canada (~4Mha)** In 2013, 2014 and 2015 the Canadian record of burned area is about ~4Mha annually. Thus Canadian forest fires is a candidate for the three elevated $NH_4^+$ events at T2015-A1 (NEEM site), as also observed in 2013 and 2014 at the T2015-A3 and T2015-A4 sites. Elevated concentration of fire tracers (levoglucosan and dehydroabietic acid) are also observed at the SE-Dome ice core in South Eastern Greenland in the same years, suggesting the signal is dispersed Greenland wide (Parvin et al., 2019).

In conclusion we find several extremes (>97.5%) in the record of $NH_4^+$ after de-trending using a five year average (Figure 7, Table 5) in our 6 traverse cores. However not all $NH_4^+$ extremes are observed in each record. The extreme $NH_4^+$ concentrations that exist through the 6 firn records are also observed in other ice core records from Greenland (NEEM, SE-dome or Summit sites) suggesting that they are fires large enough to impact a large part of the Northern hemisphere and could be used to constrain ages in shallow cores. Furthermore, we find a good correlation between our records and Canadian fire records (R=0.44, p~$10^{-4}$ Figure 8), suggesting that longer records of $NH_4^+$ from Northern Greenland, such as those from the NEEM and EastGRIP sites, can be used as a proxy for Canadian forest fires in recent times.

# 7 Conclusion

Limited sources are available on the chemical impurities deposited to the northern part of the Greenland ice sheet in recent time (e.g. Gfeller et al., 2014; Hawley et al., 2014; Vallelonga et al., 2014; Kjær et al., 2021a). We add six additional chemical proxy profiles to the large north Greenland interior in a resolution that resolves seasonal signals with the benefit that they are all retrieved and analysed by the same setup. The cores T2015-A2, T2015-A3 and T2015-A6 offer a first view on the total amount and seasonal cycles of impurities deposited to their specific central north Greenland locations. The core T2015-A1 adds to the array of cores previously drilled at the NEEM site, whilst T2015-A4 and T2015-A5 update the NEGIS core previously drilled close by at the East Greenland ice stream.

We observe a spatial gradients in conductivity and $H_2O_2$ concentration related to accumulation, while for dust (insoluble and $Ca^{2+}$) and $NH_4^+$ we do not observe significant changes in concentration between sites.

We observe similar seasonal cycles as those previously reported by others in northern Greenland, but find that our formal month definition defined mainly by $H_2O_2$ summer peaks shifts the peak deposition by up to two months compared to that found by others defining summer by a mixture of proxies. We attribute this in part to the accumulation being non evenly distributed through the year. We highlight the importance of using similar methods for constraining and dividing the year into months when aiming to investigate changes in seasonality between different ice core sites.

We observe temporal trends in the acidity and conductivity profiles related to the 1970's anthropogenic contamination of the atmosphere. In the dust fluxes we observe an increase over time, especially for the large (8.15-10 µm) and intermediate particles (2.19-8.15 µm), which could be associated with the downward trend in accumulation for the period 2000-2010 observed by (Kjær et al., 2021c) or interpreted as a sign of increased local dust in the period, as found previously by Nagatsuka et al. (2021) in the North-west, Amino et al. (2020) in a southern coastal site and by Bullard and Mockford (2018) at the west

coast. However, we note that the dust flux data and conclusions on trend remains impacted by uncertainties associated to the timescale. Despite this our inland ice core records adds to the growing evidence of a recent increase in Greenland local dust transportation.

By stacking the normalized $NH_4^+$ excess records from northern Greenland we find a good correlation with Canadian forest fires (0.49) suggesting it can be used as a proxy for specifically Canadian forest fires, more so than the individual records of $NH_4^+$. We also find several recent volcanic eruptions shown in the cores as layers in the detrended acidity and conductivity exceeding 97.5%. We find Icelandic sources for most, but note volcanic activity at the Barents Sea region could be a source for some events. Some of the assigned volcanic horizons and forest fire ammonium signals has North Greenland interior wide deposition signals that can be used to temporally constrain future firn and ice records and may further be useful for radar tracking of recent accumulation.

Despite each of the 6 firn records being analysed by similar means the spatial variation for some proxies ($Ca^{2+}$, insoluble dust and $NH_4^+$) is overwhelmed by the annual signal and additional noise from surface topography and deposition or the analytical noise in the CFA system and as a result the correlation between sites are generally not high. It is clear that more extensive investigations are essential to reduce spatial uncertainty, cancel out site specific variations and improve the representativeness of isolated locations. The two records taken close by each other at the East Greenland Ice Stream (T2015-A4 and T2015-A5 especially highlights that taking new records at a similar site as existing and interpreting changes between them as temporal changes should be done with care as even cores close by each other can look quite different in chemical composition in overlapping time periods. This study further highlights that the use of additional cores from each site is needed to constrain better depositional and analytical noise.

**Data availability**

The data sets from the 6 firn cores (T2015-A1 to T2015-A6) will be made available at www.Pangaea.de upon publication.

**Author contribution**

HK and PV collected the samples during the field season 2015. HK, PZ, PV, KHL, AS, SB analysed the firn cores by means of CFA. HK and PZ made the annual layer counting and further interpreted the chemistry data. All authors contributed to the writing of the paper.

**Acknowledgements**

The research leading to these results has received funding from the European Research Council under the European Community's Seventh Framework Programme (FP7/2007-2013) / ERC grant agreement 610055 as part of the ice2ice project. This project has received funding from the European Union's Horizon 2020 research and innovation programme under grant agreement No 820970 as part of the TiPES project.

We acknowledge EGRIP and NEEM ice core drilling projects. EGRIP is directed and organized by the Centre for Ice and Climate at the Niels Bohr Institute, University of Copenhagen. It is supported by funding agencies and institutions in Denmark (A. P. Møller Foundation, University of Copenhagen), USA (US National Science Foundation, Office of Polar Programs), Germany (Alfred Wegener Institute, Helmholtz Centre for Polar and Marine Research), Japan (National

Institute of Polar Research and Arctic Challenge for Sustainability), Norway (University of Bergen and Trond Mohn Foundation), Switzerland (Swiss National Science Foundation), France (French Polar Institute Paul-Emile Victor, Institute for Geosciences and Environmental research), Canada (University of Manitoba) and China (Chinese Academy of Sciences and Beijing Normal University). NEEM is directed and organized by the Center of Ice and Climate at the Niels Bohr Institute and US NSF, Office of Polar Programs. It is supported by funding agencies and

institutions in Belgium (FNRS-CFB and FWO), Canada (NRCan/GSC), China (CAS), Denmark (FIST), France (IPEV, CNRS/INSU, CEA and ANR), Germany (AWI), Iceland (RannIs), Japan (NIPR), Korea (KOPRI), The Netherlands (NWO/ALW), Sweden (VR), Switzerland (SNF), United Kingdom (NERC) and the USA (US NSF, Office of Polar Programs).

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
