# Peer review of "Canadian forest fires, Icelandic volcanoes and increased local dust observed in 6 shallow Greenland firn cores"

_Climate of the Past, 2021_

## Author Comment (AC1)

We would like to thank the reviewer for taking the time to consider the manuscript in details and for the great suggestions that will improve the coming version of the manuscript. Detailed replies are given in the following.

On behalf of the co-authors Helle Kjær

**Comment on cp-2021-99**

Anonymous Referee #1

**General comments**

The manuscript is focused on the analysis of the chemical records obtained by CFA from 6 shallow firn cores retrieved along the NEEM - EastGRIP Scientific Traverse. The Authors present a study of both spatial variability along the 6 sites spanning West to East

Greenland and temporal variability, after yielding an ice core chronology, basing on annual layer counting. As regarding the latter, dust concentration of sizesorted particles was used to spot possible local dust sources, free acidity and conductivity were employed to detect volcanic eruptions and a stacked ammonium record was found as a valuable proxy of forest fires in Northern America.

The paper presents an ample set of new data which can be useful to a broad community of scientists involved in recent climate reconstructions from ice core records and I find it apt to be published on Climate of the Past eventually.

We thank the reviewer for recognizing the ample work put into this manuscript as well as its usefulness for the community.

However, I find that the manuscript should go through some consistent revisions.

Some parts of the text, e.g. ice core chronology (see Specific Comments) should be better detailed and deserve a further short discussion.

We have extended this section to further elaborate the methods used to date the cores and added Figures to the supplementary that illustrates the dating better.

A general English revision is also suggested: the text is usually easy to read but sometimes sentences look as broken or dashed off hurriedly and should be rephrased. Furthermore, there are many basic format and punctuation issues which can be easily fixed.

We will carefully go through each sentence for a next version.

Here below I am listing some specific remarks to help in this process.

**Specific comments**

**Abstract**

Line 18 page 1 (and related Table 2). "Annual mean and quartiles of the...": the sentence is not immediately clear upon reading if one has not gone through the text We will revise the sentence

And Table 2 could be accompanied by a figure showing the overlap of data distributions to better appreciate it. For instance, a box and whiskers plot could be helpful, but any other solution is welcome. We thank the reviewer for this great suggestion and will add a whiskers plot of the data also presented in Table 2 and Figure 1, to better illustrate the data distributions.

**Materials and methods**

**Lines 14-16 page 4**. Melting a firn core is always a critical issue and certainly deserves some more precautions with respect to ice core sections. A melt rate of 4 cm/min sounds fine but probably even a higher rate would work. The addition of a metal coin is interesting, and I guess it is to separate the melting section from the head so that the produced water stays in contact with the firn section as little as possible, but the authors are invited to add some details about the metal coin addition. It could be better shown also in Figure S1 (here metal coin is not visible).

Indeed, the coin works to limit contact of water on the melt head with that of the snow/firn. Especially it limits the percolation into the firn by limiting contact between pores and water.

We will revise supplementary Figure 1 to illustrate better the coin and add a few sentences more on the effect of such a coin and its benefits for meting firn.

A higher melt rate could perhaps also work, but we found this low melt rate in balance with our pumping setting made the optimal amount of water available, so that water on the melt head was at all times limited to a minimum amount thus also minimizing the percolation into the firn, that is unavoidable despite the coin minimizing it.

**Section 2.1. Core chronology.** As a general remark on the section, I would invite the Authors to complete it because it lacks some details in my view.

We have chosen to split the datasets presented here into two publications. An extensive discussion of the dating, the peroxide and the subsequent accumulation is part of another paper, currently under preparation. We had hoped that this paper by now would have been finalized, but unfortunately that is not the case. We realize that this other paper is not clearly mentioned in this section. Thus we will revise the section "Core chronology" to add more details as requested.

In particular, the Authors should find a way to better show the seasonal pattern of the chosen marker, maybe making lines thinner in Figure 2 and possibly adding a figure with a close-up on a few years. It would be also interesting to read a brief discussion on the stability/loss of  $H_2O_2$  seasonality as depth increases. It cannot be appreciated from Figure 2. We will revise Figure 2 to make it a full page figure, we will further add to the supplementary for each core a plot of the Ca2+ and peroxide on a depth scale including vertical lines illustrating the individual years. We note that the combined seasonality of the marker chosen is shown also in Figure 3 top and second left plots ( $H_2O_2$  and insoluble dust).

Moreover, the Authors are invited to briefly mention the reasons why they have chosen to use only annual layer counting for the dating without using volcanic signatures of acidity and conductivity, since they have used them to study the spatial variability of volcanic eruptions in section 6.1. We have chosen to use mainly the peroxide for making the timescale, because one of the goals was to see if one could identify spatial shifts in seasonality of the other proxies. Thus by using mainly peroxide to generate the age scale (the only proxy directly related to the annual solar cycle) we hoped to see variability in the other proxies with time. Unfortunately, the signal for the low accumulation sites were not sufficient to keep the annual cycle with depth of peroxide, and in sections where H2O2 did nothave a clear annual cycle, insoluble dust seemed to be the second most stable in having a clear annual cycle.

When looking into also the reference horizons section 6.1 we have off cause also gone back and evaluated if we could argue for more or less years in some of the records to make the reference layers more consistent (e.g., between A1, A2 and A3), however we found no clear years that could be added nor removed, which would make both ammonium layers and acid layers consistent between the three cores west of the ice divide. Thus in the end we went with the simple annual layer counting, not to force layers to be consistent, but to argue in section 6.1 that we cannot rule out them being the same when only 1 yr apart.

We have reformulated in the section "chronology" as follows ". While others of the proxies analysed also show a strong annual cycle (see Figure 3) we stick to a dating based on mainly  $H_2O_2$  (or  $Ca^{2+}$ ). This is because one of the aims of the study is to investigate the seasonal cycle between sites. In addition, we note that acid horizons are commonly used to match ages between cores. However, we have chosen not to do so, as another aim for is to investigate if the acid layers in recent time do deposit between all sites. The total age of each core and the uncertainty was defined as  $\pm \frac{1}{2}$  a year for each uncertain year and can be found in Table 1.."

**Spatial variability**

**Figure 2 page 6.** As mentioned above, Figure 2 is very relevant and necessary to the manuscript but the concentration profiles from all the cores cannot be well appreciated. A simple way to make it all clearer without redrawing completely the figure is to use slightly thinner lines or maybe dashed or dotted lines for one or two cores. Any idea from the Authors in order to make it more readable is welcome.

We will revise Figure 2 to make it more readable and add additional Figures in the supplementary for each core, as well as add a whiskers plot as suggested as a supplement to Table 2.

**Lines 13-14 page 7.** Is 5 ppb a mean or median or which other reference value? Anyway, one only value as a term of comparison is not sufficient to state that "...no significant recent increase" is observed with respect to the rest of the Holocene. Please, provide a better support to this statement.

We are in this section comparing core medians with the available other published records For the Holocene we are comparing with the NEEM record (schüpbach et al., 2018, Fig 3). The 5 ppb NH4+ (Schüpbach et al, 2018, Fig3 a) is a median over the Holocene recorded of the deep NEEM record and ours from the NEEM site have a median of 5.8 ppb with 2.2 and 10.8 being the 15 and 85% quantiles respectively. Thus the two datasets are comparable and we find it fair to write the statement. We have however that this relates to the NEEM site only.

**Lines 2-4 page 8**. More than relative variability (which is lower in the NorthWest than Central and NorthEast – 15% vs. 25%, respectively), absolute values are higher, accordingly with post-depositional processes Authors mention.

We are not certain we understand the reviewer comment. Could the reviewer reformulate the concern?

We write that peroxide concentrations northwest of the ice divide is larger than east of the divide, as a result of photolysis causing loss of the deposited H2O2 at low accumulation sites (east~11 cm water equivalent accumulation annual at EastGRIP vs ~25 cm/yr at NEEM).

If the concern is that there is a larger relative variability in the 15 and 85 percentiles compared to the median west of the divide than east, we would explain that by an also more sporadic accumulation scenario east of the divide between years. However, we find it beyond scope to go into that detail in this paper. The issue and others with regards to accumulation and peroxide is discussed in another paper under preparation on accumulation and peroxide covering this same 6 firn cores.

**Lines 5-6 page 8**. Are 2 mS and 5 mS average values? Which is the associated variability? This can be important to know to evaluate if the two values are significantly different. The 15 and 85% quantiles are shown in Figure 2, as referenced in the text, but we will add a whiskers plot to further make it easier for the reader to appreciate the variation in the records.

**Seasonal cycles**

As a general remark for this section and for Figure 3, I don't find text and figure consistent: Figure 3 displays "formal season" instead of "formal month". The Figures are made based on formal months as described in the text. However, to appreciate the fact that such formal months are likely not true months, we have chosen to label the Figure with seasons only rather than months. In the discussion of section 4, we however often refer to the formal months as some proxies peak in eg. Formal month april-june, which is something between spring and summer. We acknowledge that it can make it hard to compare the text with the Figure and will therefor add also to the Figure the formal months and make the text more consistent so it refers to both seasons and formal months throughout.

Besides, seasons are reported from the right to the left (if I well interpreted) while it would be easier if they were shown in the opposite direction. I can understand that ice core records go backwards in time but in this case I find it confusing. We will reverse the direction

Also, I would replace the term "Excess" in Figure 3 with "anomaly" or, at least, would explain it well also in the caption. We use the word excess when referring to the data after removing the 5 year running average- we will clarify this in the first sentence of section 4-seasonal cycles and in the caption of the Figure showing the seasonality and stick with the word excess as this "excess" contains both the seasonal cycle, but also extreme events such as volcanic eruptions and forest fires.

We will as be suggested reverse the seasonality to go from left to right.

A higher definition would be helpful for Figure 3. We will improve the resolution

**Line 30 page 10.** It is not clear if the Authors refer to reproducibility here, how it is calculated and how "site specific noise" was evaluated. The issue of "noise" is recurring through the text, rightly so, and it deserves a more detailed discussion. We will evaluate this and other sections to be more concise about the phrase "noise". Further we will add the suggested whiskers plot and a table of correlation values between the records to better argue our claims.

**". Temporal trends**

**Line 14 page 11.** Again, the reference to "noise" should be made clearer. Do the Authors refer to the whole core or just to the most recent part? Even though median and topical quantiles are reported in Table 2, the calculation of trends and related significance would be important, in my opinion. The possible existence of trend cannot be read immediately from the Table. We will add a figure in the supplementary similar to S2 of the acid. Further we will rephrase the specific sentence; "Unfortunately, the interannual variability in the acidity record is large making it difficult to assess the temporal trend (Table 2). This is mainly a result of the measurement technique being subject to flow sensitivity (Kjær et al., 2016), but also influenced by individual peaks associated with volcanic events (discussed in section 6.1)."

**Extreme events**

I would add a mention in the section (for instance after Line 3 page 15) to the fact that other markers different from the ones analysed here can be more specific for detection and assessment of impact of volcanic eruptions (for instance, non-sea salt sulphate) as well for annual layer counting. The Authors could refer to some topical papers in the field, such as Sigl et al. (2016, CP) and Severi et al. (2012, CP). We will add as suggested "Also we note that other markers are more specific to volcanic eruptions than the ones used in this study, *e.g.* non sea salt sulphate or S isotopes."

**Line 32 page 18 – lines 1-2 page 19**. Since the Authors state (lines 9-11 page 5) that only hydrogen peroxide (with a supportive contribution of calcium) was used for dating, cannot understand now if the dating of A2 and A4 cores was tuned by using ammonium record, in the end, in order to achieve a definitive ice core chronology. It could be reasonable but it deserves a brief discussion since the time scale is basic to go on with further data interpretation. We are sorry that the text was not clear.

Indeed, only hydrogen peroxide and to some extent ca was used for the dating. However, annual layer counting is as I suspect the reviewer well knows, to some extent a subjective method, where some years can be hard to distinguish. Thus all records were annual layer counted by multiple individuals who in a few cases chose different annual peaks, allowing for some dating uncertainty as shown in Table 1. However, in the end one timescale focusing on H2O2 and calcium for the dating was chosen. Thus for most of these records as also indicated in table 1, the age is subject to some uncertainty. When investigating the peaks in ammonium, we found it surprising that the peaks between 1990-2000, looked similar in spacing but shifted. Thus here we merely test if shifting the records, the allowed +-1 yr makes the correlation to the Canadian fire index better. This shift is only invoked in this section and thus is not used in any other part of the records shown, and did not improve the correlation to the fire index either.

We have rephrased this section

"The dating for especially the eastern cores are uncertain. This allows the records to be shifted and thus as a test we shifted A2 and A4 to be one year younger, to better match the peak in 1998 and thus improve the combined proxy, however changing the dating in such way does not improve the ammonium composite ability to work as a proxy for Canadian forest fires (R-0.48,  $p10^{-4}$ , 1987-2015). "

And in the section about chronology added the information;

"Several of the other proxies analysed also show strong seasonal cycles, however as one of the aims of the study is to investigate the seasonal cycle between sites, we stick with a dating based on mainly  $H_2O_2$ , as it is the one proxy most direct related to the solar cycle. In addition, we note that acid horizons are commonly used to match ages between cores. However, we have chosen not to do so, as another aim for is to investigate if the acid layers in recent time do deposit between all sites investigated. The total age of each core and the uncertainty was defined as  $\pm \frac{1}{2}$  a year for each uncertain year and can be found in Table 1."

**Supplementary Material**

**Figure S1.** As mentioned above, please add the detail of the metal coin to the figure, since I have gathered that it is relevant to prevent the by-side effect to "backward sucking" and cannot be appreciated from the figure. Besides, a slightly higher definition for the figure would be welcome. The figure will be modified as suggested

**Technical corrections**

**Abstract**

**Ok Line 23 page 1**. I would replace "contribute" with "ascribe"

**okLine 29 page 1**. English check suggested: "peak ammonium" and "peak volcanic layers" should be corrected.

**Introduction**

**OkLine 8 page 2**. English correction: "ammonium peak concentration" should probably be "ammonium concentration maxima" or similar.

**Ok Line 12 page 2**. Add full stop and the end of the sentence (similar missing punctuation issues all through the text).

**Ok Line 15 page 2**. English change suggested: maybe "has facilitated" could be replaced by something more apt, such as "allowed obtaining".

**Methods**

**OK Lines 26-27 page 2**. Please check the format of NEEM and EastGRIP site coordinates.

**Ok Lines 5 and 6 page 2**. Check punctuation: remove an "and" and insert semicolon.

**Figure 1 page 3**. The labels of the red circles indicating the drill sites overlap one with the other and cannot be read easily. A new map have been prepared

**Ok Table 1 caption, line 7 page 3**. The reference is written in a different format from the rest of the text.

**Line 6 page 4**. In my opinion, "acid" is too vague and not corresponding to what is measured. It should be replaced by another expression, such as "acidic content", "free acidity" or just "H+" or any other apt wording. This remark holds for the whole paper (e.g. already a few lines later, line 8, again "acid"). We have changed accordingly and call it acidity when referring to the acid measure in the firn cores using the dye technique, as also done in Kjær et al. 2015 and Winstrup 2019 and acid when referring to volcanic eruptions as that can be many types of acid.

**Ok Line 10 page 4**. I guess the Authors refer to 8 pieces, each 55 cm long, please correct the expression in brackets.

**OK-only found this one place Line 17 page 4**. Please correct ammonium formula using superscript. Check carefully these format issues all through the text.

**Ok Line 20 page 4**. I would replace "in sufficient resolution" with "with sufficient resolution".

**OK Line 22 page 4.** I would write "it is produced" adding a verb. Otherwise, please rephrase.

**Ok-rephrased Line 27 page 4**. "Sufficiently high enough" contains a repetition, I find.

**Ok Line 3 page 5**. Please use the same shortened name for the same core (e.g. 2015T-A6 or T2015-A6).

**Lines 6-11 page 5**. There is probably an issue with tense of verbs; please choose past tense (as mostly used in the rest of the text) or present.

**Corrected to 15 and 85 both places.Table 2 caption page 7**. It is quite peculiar that you use 15th and 85th percentile here while you use 16th and 84th percentile in Figure 3; I don't think it changes the result, of course, am just curious to know.

**Spatial variability**

**OK Figure 2 caption page 6.** As remarked earlier, I would replace the expression "acid", here and through all the text.

**Ok Table 2 (page 6 and 7).** Please, check the format of the analysed parameters (namely superscripts and symbol for "micro").

**OK Table 2 caption (page 6 and 7).** I would add some details for the unit of measurement for dust in the Table or in the caption. Is it "#" referring to the total number of particles or to one particular size range?

**OK Line 10 page 6.** They are not "estimates", actually; I would use the word "measurements".

**Rephrased Line 11 page 7.** "Lower estimate": what do the Authors mean with it? The minimum value? A small percentile?

**OK Line 15 page 7.** Please, add the right symbol (±).

**Line 20 page 7.** "Counts mL-1" is an unit of measurement for a signal, not for a concentration, which I find it more correct, to estimate a noise (signal is highly variable among different instruments, also in the case of dust measurements, I believe).

Unfortunately we do not understand this reviewer comment, could the reviewer please re-iterate the concern. The dust is measured in counts of particles (1-10 Um) per mL?

**Seasonal cycles**

**Ok Line 5 page 10 (also line 18 page 18).** Please add brackets for publishing year for Gfeller et al. (2014).

OK Line 8 page 10. As above.

Temporal trends

**Is present** McIlhattan, E. A., Pettersen, C., Wood, N. B., and L'Ecuyer, T. S.: Satellite observations of snowfall regimes over the Greenland Ice Sheet, The Cryosphere, 14, 4379–4404, https://doi.org/10.5194/tc-14-4379-2020, 2020 **Line 19 page 11.** The reference does not appear in the Reference list.

**Ok Line 21 page 11.** Please, correct of format of "micro", also later in the section

**Ok Line 29 page 11.** "assuming all spheres were perfectly round": would rephrase f.i. "assuming all particles are perfectly round".

**Rephrased Lines 4-5 page 12**. Please, rewrite the sentence starting with "Thus"; it appears to be broken.

"The largest particles (>10.5  $\mu$ m) are omitted from further analysis as they are subject to poor statistics and the smallest sizes (<1.25  $\mu$ m) as well as they are noisy"

**Rephrased Line 6 page 12.** I would complete the sentence this way: "...parting the data set this way..."

"We find that by parting the dust data this way we have 12-28% of the total dust in the small range..."

**OK Table 3 page 13.** check format (width of the first column, superscript in header of the second column, ...)

**Extreme events**

**OK Line 3 page 16 and line 5 page 17.** Check format (superscript in km3).

**OK -found only this and one other incidence Line 17 and line 31 page 16.** Please, do not use the shortened expression "1986 Nov" and similar in the text

**Rephrased Line 5 page 17**. After "...eruption signal" the sentence is not clear, please rewrite.

**Ok Line 23 page 18.** Naming the sites located west of the ice divide would help the reader who is not extremely familiar with Greenland morphology.

**Ok Lines 28-29 page 18.** Please check the format of p value.

We will redraw the lines **Figure 6 page 19**. Dotted lines for the fire records are not well visible.

**Ok Line 5 page 19.** ">97.5% of full records": I assume the Authors refer to the 97.5th of each full record but it would be useful if they report it explicitly.

OK Line 18 page 20. No capital letter is needed for "levoglucosan"

**OK Line 19 page 20.** I believe "high concentration values" or "concentration peaks" are missing in the sentence. Same at line 10 for dehydroabietic acid and **line 14** for fire tracers.

**OK Line 13 page 20.** NEEM is with capital letters.

We will add this information. **Line 21 page 20.** I am sure this correlation coefficient (is it R or R2, by the way?) is highly significant but the Authors could report the associated significance and the number of data as well.

**Conclusions**

OK Line 7 page 21. Please correct the symbols of "micro".

**OK Lines 9-10 page 21.** Please, correct the format of publication year for Nagatsuka et al. and Amino et al. Again, the sentence starting with "Thus" appears to be broken, please rephrase.

**Data availability**

OK Please check punctuation and core names.

**References**

**Yes published Lines 24-26 page 24.** This paper should be published now and not on TCD anymore; please, update.

**Supplementary Material**

**OK Line 3 page 3**. Please correct format for hydrogen peroxide (subscripts)

---

## Author Response (AR1)

We would like to thank the reviewers for taking the time to consider the manuscript in details and for the great suggestions that improved this version of the manuscript presenting an ample set of new data.

Especially the idea of a whiskers plot (new Figure 3) and to show the individual records (Supplementary S2-S7) were very beneficial for the revision. In addition, correlation coefficients between monthly and annual records for all proxies are added to the supplementary and now discussed also in the text.

Further some of the comments and new Figures prompted a close look at the age scale, which unfortunately we fund was corrupted for some of the records, thus for this version all cores have been re-dated and all Figures and numbers are updated accordingly.

Major updates include:

1) The age scale for all records have been remade, and thus also all Figures and Tables
2) In the supplementary Figures S2 to S7 showing the individual cores on a depth scale has been added.
3) All Figures have been changed to a colorblind friendly scale and resolution has been increased
4) Whiskers plot have been added (Figure 3), and old Table 2 (now Table S4) has been moved to the supplementary as the two depict the same data.
5) Correlation data (and p-values) between the proxies in the individual records are added for monthly and annual resolved data in the supplementary section S3 and in the end of section 3 these are discussed.
6) In Figure 4 (old Figure 3) showing seasonality the 15 and 85 percentiles are shown as dashed lines to make it easier to see the individual records
7) Figure 6 and 7 the 97.5 % has been made thicker and thus more visible.
8) A major revision of the text guided by the comments from the reviewers have been undertaken.

Detailed replies are given in the following.

On behalf of the co-authors
Helle Kjær

**Comment on cp-2021-99**
Anonymous Referee #1

**General comments**

The manuscript is focused on the analysis of the chemical records obtained by CFA from 6 shallow firn cores retrieved along the NEEM - EastGRIP Scientific Traverse. The Authors present a study of both spatial variability along the 6 sites spanning West to East
Greenland and temporal variability, after yielding an ice core chronology, basing on annual layer counting. As regarding the latter, dust concentration of size-sorted particles was used to spot possible local dust sources, free acidity and conductivity were employed to detect volcanic eruptions and a stacked ammonium record was found as a valuable proxy of forest fires in Northern America.

The paper presents an ample set of new data which can be useful to a broad community of scientists involved in recent climate reconstructions from ice core records and I find it apt to be published on Climate of the Past eventually.

We thank the reviewer for recognizing the ample work put into this manuscript as well as its usefulness for the community.

However, I find that the manuscript should go through some consistent revisions.

Some parts of the text, e.g. ice core chronology (see Specific Comments) should be better detailed and deserve a further short discussion.

We have extended this section to further elaborate the methods used to date the cores and added Figures to the supplementary (Figures S2-S7) that illustrates the dating better.

A general English revision is also suggested: the text is usually easy to read but sometimes sentences look as broken or dashed off hurriedly and should be rephrased. Furthermore, there are many basic format and punctuation issues which can be easily fixed.

We have done our best and gone carefully through the manuscript.

Here below I am listing some specific remarks to help in this process.

**Specific comments**

**Abstract**

**Line 18 page 1 (and related Table 2)**. "Annual mean and quartiles of the…": the sentence is not immediately clear upon reading if one has not gone through the text The abstract has been completely rewritten to make it more concise

And Table 2 could be accompanied by a figure showing the overlap of data distributions to better appreciate it. For instance, a box and whiskers plot could be helpful, but any other solution is welcome. We thank the reviewer for this great

suggestion and we have added a whiskers plot (new Figure 3) of the data and removed previous Table 2 to the supplementary (now Table S4).

**▪ Materials and methods**

**Lines 14-16 page 4**. Melting a firn core is always a critical issue and certainly deserves some more precautions with respect to ice core sections. A melt rate of 4 cm/min sounds fine but probably even a higher rate would work. The addition of a metal coin is interesting, and I guess it is to separate the melting section from the head so that the produced water stays in contact with the firn section as little as possible, but the authors are invited to add some details about the metal coin addition. It could be better shown also in Figure S1 (here metal coin is not visible).

Indeed, the coin works to limit contact of water on the melt head with that of the snow/firn. Especially it limits the percolation into the firn by limiting contact between pores and water.

We have added a coin to Figure S1 and adde a few sentences more on the effect of such a coin and its benefits for meting firn.The section now reads

"Driven by capillary forces the melt water percolates from the CFA melt head into the firn core above the melt head dispersing the signal. This was mitigated by adding a metal (97 % Cu, 2,5 % Zn) coin to the melt head to limit contact between any excess meltwater on the melt head and the firn core. In addition, such excess water that could be sucked up into the firn was limited by carefully adjusting melt head temperature relative to pump speeds carrying the water away. With these modifications the level of water percolating into the firn from the melt head was limited to <1cm. Melt rate was kept at ~4 cm/min which resulted in the final depth resolution of the ions measured being <2 cm ($H^+_{dye}$, $NH^{4+}$, $H_2O_2$, $Ca^{2+}$), while for the conductivity and dust with shorter step-change response times (time it takes to go from a level of 5% to 95% of a concentration) a depth resolution of 8 mm was achieved. We note that the accumulation at the sites vary between 12 and 23 cm w.eq $a^{-1}$ and thus annual signals are resolved with the achieved resolution.""

A higher melt rate could perhaps also work, but we found this low melt rate in balance with our pumping setting made the optimal amount of water available, so that water on the melt head was at all times limited to a minimum amount thus also minimizing the percolation into the firn, that is unavoidable despite the coin minimizing it.

**Section 2.1. Core chronology.** As a general remark on the section, I would invite the Authors to complete it because it lacks some details in my view.

We have chosen to split the datasets presented here into two publications. An extensive discussion of the dating, the peroxide and the subsequent accumulation is part of another paper, currently under preparation. We had hoped that this paper by now would have been finalized, but unfortunately that is not the case. We realize that this other paper is not clearly mentioned in this section. Thus we have revised the section "Core chronology" and added more details as requested.

In particular, the Authors should find a way to better show the seasonal pattern of the chosen marker, maybe making lines thinner in Figure 2 and possibly adding a figure with a close-up on a few years. It would be also interesting to read a brief discussion on the stability/loss of $H_2O_2$ seasonality as depth increases. It cannot be appreciated from Figure 2. We have revised Figure 2 to make it a full page figure, we will further add to the supplementary for each core a plot of all proxies on a depth scale including vertical lines illustrating the individual years (Supplementary Figure S2-S7). We note that the combined seasonality of the marker chosen is shown also in Figure 4 ($H_2O_2$ and insoluble dust).

Moreover, the Authors are invited to briefly mention the reasons why they have chosen to use only annual layer counting for the dating without using volcanic signatures of acidity and conductivity, since they have used them to study the spatial variability of volcanic eruptions in section 6.1. We have chosen to use mainly the peroxide for making the timescale, because one of the goals was to see if one could identify spatial shifts in seasonality of the other proxies. Thus by using mainly peroxide to generate the age scale (the only proxy directly related to the annual solar cycle) we hoped to see variability in the other proxies with time. Unfortunately, the signal for the low accumulation sites were not sufficient to keep the annual cycle with depth of peroxide, and in sections where H2O2 did not have a clear annual cycle, insoluble dust seemed to be the second most stable in having a clear annual cycle. This has been added in the text.

When looking into also the reference horizons section 6.1 we have off cause also gone back and evaluated if we could argue for more or less years in some of the records to make the reference layers more consistent (e.g., between A1, A2 and A3). In this process we realized that the dating file used for T2015-A5 ad T2015-A4 had been corrupted and thus in the end we redid the annual layer counting on all cores. Again strictly relying on H2O2 and Ca2+.

We have reformulated in the section "chronology" as follows *"We rely on the strong seasonal pattern of $H_2O_2$ (Figure S2 to S7, top) to constrain the age of the 6 shallow cores (Table 1), where we assign the summer maxima of $H_2O_2$ to solar solstice (June). At the low accumulation sites where $H_2O_2$ seasonality was not well resolved; T2015-A4, T2015-A5, and T2015-A6 the seasonality in $Ca^{2+}$ (Figure S2-S7, second topmost) was used to further constrain the firn core chronologies. Despite the fact that others of the proxies analysed also show a strong annual cycle (see Figure 2, and also Figure 4) we stick to an age scale based on just $H_2O_2$ (or $Ca^{2+}$). This is because one of the aims of the study is to investigate the seasonal cycle between sites. In addition, we note that acid horizons are commonly used to match ages between cores. However, we have chosen not to do so, as another aim for is to investigate which of the extreme acid layers in recent time that can be used to constrain ages between sites. The total age of each core and the uncertainty was defined as ± ½ a year for each uncertain year and can be found in Table 1. We then use the age-depth relationship from the $H_2O_2$ peaks to interpolate the depth series into a time series using a constant accumulation assumption. Accumulation from the GC network at NEEM suggests that a fairly equal summer to winter ratio (Gfeller et al., 2014) and thus we stick with a simple constant accumulation scenario (Gfeller et al., 2014; Kjær et al., 2013). We could have used re-analysis accumulation data to constrain the monthly accumulation, but even high-resolution weather re-analysis performs poorly on the central ice sheet (Kjær et al., 2021c)"*

**Figure 2 page 6.** As mentioned above, Figure 2 is very relevant and necessary to the manuscript but the concentration profiles from all the cores cannot be well appreciated. A simple way to make it all clearer without redrawing completely the figure is to use slightly thinner lines or maybe dashed or dotted lines for one or two cores. Any idea from the Authors in order to make it more readable is welcome.

We have modified Figure 2 with new colors to make it better visible for all. Also we have made it full page and enhanced the resolution. In addition Figure S2-S7 adds the individual records on a depth scale.

**Lines 13-14 page 7.** Is 5 ppb a mean or median or which other reference value? Anyway, one only value as a term of comparison is not sufficient to state that "…no significant recent increase" is observed with respect to the rest of the Holocene. Please, provide a better support to this statement.

We are in this section comparing core medians with the available other published records For the Holocene we are comparing with the NEEM record (schüpbach et al., 2018, Fig 3). The 5 ppb $NH_4^+$ (Schüpbach et al, 2018, Fig3 a) is a median over the Holocene recorded of the deep NEEM record and ours from the NEEM site have a median of 5.8 ppb with 2.2 and 10.8 being the 15 and 85% quantiles respectively. Thus the two datasets are comparable and we find it fair to write the statement. We have however added that this relates to the NEEM site only.

**Lines 2-4 page 8**. More than relative variability (which is lower in the NorthWest than Central and NorthEast – 15% vs. 25%, respectively), absolute values are higher, accordingly with post-depositional processes Authors mention.

We are not certain we understand the reviewer comment. Could the reviewer reformulate the concern?

We write that peroxide concentrations northwest of the ice divide is larger than east of the divide, as a result of photolysis causing loss of the deposited H2O2 at low accumulation sites (east~12 cm water equivalent accumulation annual at EastGRIP vs ~23 cm/yr at NEEM).

If the concern is that there is a larger relative variability in the 15 and 85 percentiles compared to the median west of the divide than east, we would explain that by an also more sporadic accumulation scenario east of the divide between years. However, we find it beyond scope to go into that detail in this paper. The issue and others with regards to accumulation and peroxide is discussed in another paper under preparation on accumulation and peroxide covering this same 6 firn cores.

**Lines 5-6 page 8**. Are 2 mS and 5 mS average values? Which is the associated variability? This can be important to know to evaluate if the two values are significantly different. A whiskers plot (new Figure 3) has been added to better show the data and we note that the 15-85% was given also in no Table S4.

■ **Seasonal cycles**

As a general remark for this section and for Figure 3, I don't find text and figure consistent: Figure 3 displays "formal season" instead of "formal month". This relates to now Figure 4. The Figures are made based on formal months as described in the text. However, to appreciate the fact that such formal months are likely not true months, we had chosen to label the Figure with seasons only rather than months. In the discussion of section 4, we however often refer to the formal months as some proxies peak in eg. Formal month april-june, which is

something between spring and summer. We acknowledge that it can make it hard to compare the text with the Figure and have therefor used instead formal months and made the text more consistent so it refers to formal months throughout. Further the seasonal Figure has been modified to make the 15 an 85 percentiles easier to distinguish.

Besides, seasons are reported from the right to the left (if I well interpreted) while it would be easier if they were shown in the opposite direction. I can understand that ice core records go backwards in time but in this case I find it confusing. Also, I would replace the term "Excess" in Figure 3 with "anomaly" or, at least, would explain it well also in the caption. We use the word excess when referring to the data after removing the 5 year running average- we will clarify this in the first sentence of section 4-seasonal cycles and in the caption of the Figure 4 showing the seasonality and stick with the word excess as this "excess" contains both the seasonal cycle, but also extreme events such as volcanic eruptions and forest fires.

We will as be suggested reverse the seasonality to go from left to right.

A higher definition would be helpful for Figure 3. This relates to current Figure 4. We have improved the resolution for all Figures

**Line 30 page 10.** It is not clear if the Authors refer to reproducibility here, how it is calculated and how "site specific noise" was evaluated. The issue of "noise" is recurring through the text, rightly so, and it deserves a more detailed discussion. We have changing the wording here and elsewhere to be more precise on what kind of noise we are talking about, thus we now use words as "spatial variation" to be more concise

Further we have added the suggested whiskers plot (Figure 3) and a table of correlation values supplementary section S3 between the individual proxies to better argue our claims in this section, which now has also some lines specifically on the correlation between monthly vs annual records.

**"▪ Temporal trends**

**Line 14 page 11.** Again, the reference to "noise" should be made clearer. Do the Authors refer to the whole core or just to the most recent part? Even though median and topical quantiles are reported in Table 2, the calculation of trends and related significance would be important, in my opinion. The possible existence of trend cannot be read immediately from the Table. We have added also the 5 year trends in the acid in Figure S8 in the supplementary. The discussion of noise has been move to section 3 and now reads "Finally, we note that high resolution records, as in this study contain variations related not only to the climatology, but also to the analytical setup (eg. smoothing for the different CFA systems) and/or site specific noise and this noise limits the records capability to resolve spatial gradients between the firn records. Site specific noise is related to the local precipitation patterns, which can be disturbed by wind causing the formation of dunes, sastrugis or crust layers. These features mix up already deposited snow especially if precipitation is very event based. Melt layers at sites experiencing higher temperatures and ablation can also redistribute the deposited ions in the snow pack (Laepple et al., 2016; Gfeller et al., 2014)"

- **Extreme events**

I would add a mention in the section (for instance after Line 3 page 15) to the fact that other markers different from the ones analysed here can be more specific for detection and assessment of impact of volcanic eruptions (for instance, non-sea salt sulphate) as well for annual layer counting. The Authors could refer to some topical papers in the field, such as Sigl et al. (2016, CP) and Severi et al. (2012, CP). We will add as suggested "Also we note that other markers are more specific to volcanic eruptions than the ones used in this study, *e.g.* non sea salt sulphate or S isotopes  (Severi et al., 2012; Sigl et al., 2016a; Mayewski et al., 1990; Lin et al., 2022; Crick et al., 2021).""

**Line 32 page 18 – lines 1-2 page 19**. Since the Authors state (lines 9-11 page 5) that only hydrogen peroxide (with a supportive contribution of calcium) was used for dating, cannot understand now if the dating of A2 and A4 cores was tuned by using ammonium record, in the end, in order to achieve a definitive ice core chronology. It could be reasonable but it deserves a brief discussion since the time scale is basic to go on with further data interpretation. We are sorry that the text was not clear.

Indeed, only hydrogen peroxide and to some extent Ca2+ was used for the dating. However, annual layer counting is as I suspect the reviewer well knows, to some extent a subjective method, where some years can be hard to distinguish. Thus all records were annual layer counted by multiple individuals who in a few cases chose different annual peaks, allowing for some dating uncertainty as shown in Table 1. However, in the end one timescale focusing on H2O2 and calcium for the dating was chosen. Thus for most of these records as also indicated in table 1, the age is subject to some uncertainty. When investigating the peaks in ammonium, we found it surprising that the peaks between 1990-2000, looked similar in spacing but shifted.

We for this review had a closer look at the agescale file and found that unfortunately it had been corrupted, Thus we have redated the cores again and the new timescale the ammonium records while still not used for counting the years look much more coherent, whch gives us confidence that the current dating is accurate.

And in the section about chronology added the information;

"Despite the fact that others of the proxies analysed also show a strong annual cycle (see Figure 2, and also Figure 4) we stick to an age scale based on just $H_2O_2$ (or $Ca^{2+}$). This is because one of the aims of the study is to investigate the seasonal cycle between sites. In addition, we note that acid horizons are commonly used to match ages between cores. However, we have chosen not to do so, as another aim for is to investigate which of the extreme acid layers in recent time that can be used to constrain ages between sites. The total age of each core and the uncertainty was defined as ± ½ a year for each uncertain year and can be found in Table 1".

Supplementary Material

**Figure S1.** As mentioned above, please add the detail of the metal coin to the figure, since I have gathered that it is relevant to prevent the by-side effect to "backward sucking" and cannot be appreciated from the figure. Besides, a slightly higher definition for the figure would be welcome.The figure is modified as suggested and additional information on the coin is added to the main text.

**Technical corrections**

**Abstract**

**Ok** **Line 23 page 1**. I would replace "contribute" with "ascribe"

**ok** **Line 29 page 1**. English check suggested: "peak ammonium" and "peak volcanic layers" should be corrected.

**Introduction**

**Ok** **Line 8 page 2**. English correction: "ammonium peak concentration" should probably be "ammonium concentration maxima" or similar.

**Ok** **Line 12 page 2**. Add full stop and the end of the sentence (similar missing punctuation issues all through the text).

**Ok** **Line 15 page 2**. English change suggested: maybe "has facilitated" could be replaced by something more apt, such as "allowed obtaining".

**Methods**

**OK** **Lines 26-27 page 2**. Please check the format of NEEM and EastGRIP site coordinates.
**Ok** **Lines 5 and 6 page 2**. Check punctuation: remove an "and" and insert semicolon.
**Figure 1 page 3**. The labels of the red circles indicating the drill sites overlap one with the other and cannot be read easily. A new map have been prepared

**Ok** **Table 1 caption, line 7 page 3**. The reference is written in a different format from the rest of the text.

**Line 6 page 4**. In my opinion, "acid" is too vague and not corresponding to what is measured. It should be replaced by another expression, such as "acidic content", "free acidity" or just "H+" or any other apt wording. This remark holds for the whole paper (e.g. already a few lines later, line 8, again "acid").
We have changed accordingly and call it $H^+_{dye}$ when referring to the acid measure in the firn cores using the dye technique, as also done in Kjær et al. 2015 and Winstrup 2019 and acid when referring to volcanic eruptions as that can be many types of acid.

**Ok** **Line 10 page 4**. I guess the Authors refer to 8 pieces, each 55 cm long, please correct the expression in brackets.

**OK** **Line 17 page 4**. Please correct ammonium formula using superscript. Check carefully these format issues all through the text.

**Ok** **Line 20 page 4**. I would replace "in sufficient resolution" with "with sufficient resolution".

**OK** **Line 22 page 4.** I would write "it is produced" adding a verb. Otherwise, please rephrase.
**Ok-rephrased** **Line 27 page 4**. "Sufficiently high enough" contains a repetition, I find.
**Ok** **Line 3 page 5**. Please use the same shortened name for the same core (e.g. 2015T-A6 or T2015-A6).

**Lines 6-11 page 5**. There is probably an issue with tense of verbs; please choose past tense (as mostly used in the rest of the text) or present.

**Corrected to 15 and 85 both places.Table 2 caption page 7**. It is quite peculiar that you use 15th and 85th percentile here while you use 16th and 84th percentile in Figure 3; I don't think it changes the result, of course, am just curious to know.

- **Spatial variability**

**OK Figure 2 caption page 6.** As remarked earlier, I would replace the expression "acid", here and through all the text.

**Ok Table 2 (page 6 and 7).** Please, check the format of the analysed parameters (namely superscripts and symbol for "micro").

**OK Table 2 caption (page 6 and 7).** I would add some details for the unit of measurement for dust in the Table or in the caption. Is it "#" referring to the total number of particles or to one particular size range?

**OK Line 10 page 6.** They are not "estimates", actually; I would use the word "measurements".

**Rephrased Line 11 page 7.** "Lower estimate": what do the Authors mean with it? The minimum value? A small percentile?

**OK Line 15 page 7.** Please, add the right symbol (±).

**Line 20 page 7.** "Counts mL$^{-1}$" is an unit of measurement for a signal, not for a concentration, which I find it more correct, to estimate a noise (signal is highly variable among different instruments, also in the case of dust measurements, I believe).

Unfortunately we do not understand this reviewer comment, could the reviewer please re-iterate the concern. The dust is measured in counts of particles (1-10 Um) per mL?

- **Seasonal cycles**

**Ok Line 5 page 10 (also line 18 page 18).** Please add brackets for publishing year for Gfeller et al. (2014).

**OK Line 8 page 10.** As above.

- **Temporal trends**

**Line 19 page 11.** The reference does not appear in the Reference list.

**It is the following;** McIlhattan, E. A., Pettersen, C., Wood, N. B., and L'Ecuyer, T. S.: Satellite observations of snowfall regimes over the Greenland Ice Sheet, The Cryosphere, 14, 4379–4404, https://doi.org/10.5194/tc-14-4379-2020, 2020

**Ok Line 21 page 11.** Please, correct of format of "micro", also later in the section

**Ok Line 29 page 11.** "assuming all spheres were perfectly round": would rephrase f.i. "assuming all particles are perfectly round".

**Rephrased Lines 4-5 page 12**. Please, rewrite the sentence starting with "Thus"; it appears to be broken.

"The largest particles (>10.5 µm) are omitted from further analysis as they are subject to poor statistics and the smallest sizes (<1.25 µm) as well as they are noisy"

**Rephrased Line 6 page 12**. I would complete the sentence this way: "…parting the data set this way…"

"We find that by parting the dust data this way we have 12-28% of the total dust in the small range…"

**OK Table 3 page 13.** check format (width of the first column, superscript in header of the second column, …)

▪ **Extreme events**

**OK Line 3 page 16 and line 5 page 17.** Check format (superscript in $km^3$).

**OK  Line 17 and line 31 page 16.** Please, do not use the shortened expression "1986 Nov" and similar in the text

**Rephrased Line 5 page 17**. After "…eruption signal" the sentence is not clear, please rewrite.

**Ok Line 23 page 18.** Naming the sites located west of the ice divide would help the reader who is not extremely familiar with Greenland morphology.

**Ok Lines 28-29 page 18.** Please check the format of p value.

We will redraw the lines **Figure 6 page 19**. Dotted lines for the fire records are not well visible.

**Ok Line 5 page 19.** ">97.5% of full records": I assume the Authors refer to the $97.5^{th}$ of each full record but it would be useful if they report it explicitly.

**OK Line 18 page 20.** No capital letter is needed for "levoglucosan"

**OK Line 19 page 20.** I believe "high concentration values" or "concentration peaks" are missing in the sentence. Same at line 10 for dehydroabietic acid and **line 14** for fire tracers.

**OK Line 13 page 20.** NEEM is with capital letters.

We will add this information. **Line 21 page 20.** I am sure this correlation coefficient (is it R or $R^2$, by the way?) is highly significant but the Authors could report the associated significance and the number of data as well.

**Conclusions**

**OK Line 7 page 21.** Please correct the symbols of "micro".

**OK Lines 9-10 page 21.** Please, correct the format of publication year for Nagatsuka et al. and Amino et al. Again, the sentence starting with "Thus" appears to be broken, please rephrase.

**Data availability**

OK Please check punctuation and core names.

**References**

**Yes published Lines 24-26 page 24.** This paper should be published now and not on TCD anymore; please, update.

**Supplementary Material**

**OK Line 3 page 3**. Please correct format for hydrogen peroxide (subscripts)

**Comment on cp-2021-99**

Anonymous Referee #2
* * *
Referee comment on "NEEM to EastGRIP Traverse – spatial variability, seasonality, extreme events and trends in common ice core proxies over the past decades" by Helle Astrid Kjær et al., Clim. Past Discuss., https://doi.org/10.5194/cp-2021-99-RC2, 2021
* * *
Review of manuscript cp-2021-99:

"NEEM to EastGRIP Traverse - spatial variability, seasonality, extreme events and trends in common ice core proxies over the past decades" by Helle Astrid Kjær et al.

This study presents new impurity data from six firn cores taken along a spatial transect in North Greenland and measured with a Continuous Flow Analysis system. The data are investigated in regard to their mean seasonal cycle, their temporal trends and their usability for indicating volcanic events and as a forest fire proxy. While the results could be of interest to the ice core and proxy community and hence to the Climate of the Past readers, I rate this paper to not sufficiently well communicate the novelty of the results and its overall presentation quality to be poor. In brief, I recommend the paper to be rejected.

We thank the reviewer for recognizing the importance of the records presented here to the community and are sorry that the reviewer finds the communication of the results not sufficient. We are hopeful that with the great comments and suggestions from both reviewers a second version of the manuscript including significant modifications will succeed to communicate the novel results from these 6 firn cores.

**Major comments**

**Overall structure and writing.** In my opinion, the text seems carelessly assembled and is poorly written. Many parts and sections lack a clear structure, most notably the abstract and the various results sections. Especially regarding the latter, there is no clear distinction between the presentation of the study's results and their discussion. While the journal offers the possibility of a combined results and discussion section, I find it unusual that the authors chose to start many sections with some kind of short literature review before they actually present their own new results.

We thank you for these comments and have removed large parts of the introduction from Section 4, 5 and 6 to the introduction. But we have chosen to keep the results and discussion part together, as we deliberately chose this format because the proxies inform on very different parts of the climate system. Thus we find the reader could be from a variety of backgrounds perhaps interested in just one particular proxy. The way we have written section 4, 5 and 6 it would be easy for the reader to identify which part of the paper is of their main interest.

Additionally, results are very often stated or mentioned without any clear reference to a figure or table, which makes it difficult for the reader to retrace and verify a particular result.

We have added a number of references to Figures and Tables throughout. In addition, a new Figure 3 showing the statistics of the dataset to the main text and Figures showing the raw data on a depth scale to the supplementary.

Regarding the writing, the text suffers from frequent grammatical mistakes and "orphan sentences" which lack a subject or syntactically just peter out.

We will work on improving the English level throughout. However, it remains as always a challenge to communicate short and concise in a non-native language.

**Figure quality.** The figure quality is very poor overall. The resolution is too low, making the graphics grainy already at standard zoom, the labeling is faint and too small, and the line plots are thin and are using color scales that are very hard to distinguish and are even indistinguishable for color-blind people. I would strongly recommend the authors to study how to produce higher-quality graphics from the computer program in use, either by using scalable vector graphics or by using a sufficiently high dpi value for raster graphics. In addition, color scales which are legible for color-blind people and sufficiently distinct both for on-screen viewing as well as printing can be looked up on resources such as https://colorbrewer2.org. We have modified all plots. We have enhanced resolution and added a new color scheme. We note that while the Figures are added to the word file used to generate the text for review we do have them in .eps also for the eventual published online version.

**Local deposition noise.** Local deposition noise, and also the noise from intermittent precipitation, is an important issue but not treated appropriately in this study. It is either mentioned somewhat unmotivated, as is the case for example on P10 Line 29, or only briefly referred to at instances scattered throughout the text. In the recent years quite some literature was published on these topics, both for Greenland and Antarctica, which could be used to put the current data into context. While the available data might be a bit limited for this purpose, one could perform at least some statistical investigations, e.g., looking at the correlations between profiles at seasonal and annual resolution to see if there is any common signal among the cores along the traverse, depending on the impurity species.

We are thankfull for this suggestion and have added the suggested correlations between sites in the Supplementary and added a discussion on it in the main text section 3. Further the text has been rewritten throughout to try to avoid the word noise and rather use other more concise phrasings.

If the reviewer knows of additional works, we should compare with please do let us know the paper references.

**Trends (Section 5).** In general, it is very difficult to follow from where you derive your results and conclusions about the various trends. Overall, the paper would benefit from showing additionally a plot with the annual mean time series for the individual impurities, maybe even showing only stacked annual mean time series from averaging across the firn cores in favor of a clear presentation. Then, clearly stating the results from linear regressions on the data, including slope

uncertainty and p values, might help creating a concise picture on the overall trends.

To show better the individual records we have added Figure S2-S7, which show the individual records on a depthscale. We find that in combination with Figure 2, the individual data is now easy to see.

The temporal trends discussed for the acid/conductivity are not linear with time. We found that Figure 2 of the conductivity is enough to show the trend of a return form a 1970's high. In addition, we have in the supplementary already added a 5 year running average for each core of the conductivity (old Figure S2, now Figure S8) showing clearly the decline from the 1970's. The acid as also mentioned in the text is subject to more annual variability and to measurement noise, but we will add in the supplementary a Figure similar to S2 of the acid.

In case of insoluble dust fluxes, you do use annual mean time series, but for unknown reasons they are relegated to the supplement, and the trend results mentioned in the text are hard to verify by looking at Fig. S3. Maybe a logarithmic y axis scaling and adding the trend lines to the plot might help here. We have moved Figure 5 (previous version S3) into the main document and chosen to present the trends and p-values of the dust in Table 2. In the suplementary we have in addition added a Figure resembling Figure 5 but with trend lines and only for the period discussed. As well as tables showing the dust flux trend in percent (S8) and a Table similar to Table 2 for the longer period where there is overlap between cores(1988 onwards).

**Section 6.** Overall, these sections are overly detailed, making it hard for the reader to grasp the main conclusions you want to convey here. One idea could be to put all of the results concerning the determined extreme and volcanic events, and the possible sources thereof, into a table, maybe also giving some indication for how certain you can be on relating a specific event in the records to a known eruption or other source. Then, the text could be significantly shortened to concisely present the main findings and conclusions from this table, which could make it much clearer for the reader how the new data can possibly advance our knowledge on the mentioned topics. We would like to thank the reviewer for this excellent suggestion. We have added Tables (Table 4 and 5) providing overview of the volcanic eruptions/forest fires observed in this study and other records where they are identified and added headers to the text to make it easier to grasp.

**Minor comments**

**General.** Frequently, the term "excess" is used to refer to specific data series, however, what this terminology means is nowhere explained. This is problematic since it is firstly not a common terminology for data series, and secondly it might be confused with the quantity of "deuterium excess" commonly measured on firn and ice cores. From what I understand, your usage of "excess" refers to either the deviations from the mean of the seasonal cycle data (Fig. 3), which more commonly would be referred to as "anomalies", or to the residuals after subtracting a five-year running mean from the data series (e.g., Fig. 4). I would suggest to adopt a more appropriate terminology or to clearly define your usage of the term "excess" in the Methods. We thank the reviewer for this comment. Our usage of excess throughout refers to what is left after removing the running 5 year mean, we use the word excess as what we have contains the seasonal cycle

as well as extreme events and any measurement and site specific noise. Thus we find that neither the word residuals or anomalies are quite adequate for what we have left. We will define the term and use at first occurrence in the manuscript.

In section 2 we write "To investigate the seasonality in the proxies we first removed the five year running average and we use the term excess for the remainder. The years were split further into 12 months of equal accumulation using the formal month definition (Gfeller et al., 2014; Kjær et al., 2016)."

And in section 4 we highlight again what we determine as excess "We remove the five year running average and in the excess investigate the seasonality by formal month (Figure 4). Thus the average seasonal cycle of excess concentration after removing the five-year mean contains also extreme events such as forest fires and volcanic horizons, which is discussed in more detail in section 5-Temporal trends.

"

**Title.** I find the title too long and too general, merely listing key words rather than naming the key essence of the paper. In addition, the title should not have a full stop (I am referring to the pdf version here). Should be gradients if this title is used.

We struggle to come up with a good and short title that is still descriptive. We went with **"**Canadian forest fires, Icelandic volcanoes and increased local dust observed in 6 shallow Greenland firn cores**"**

**Abstract.** In my opinion, the abstract could be significantly shortened to convey only a brief introduction as well as the key messages and results of the study. There are several unnecessary filler sentences, e.g., "The temporal variability of the records is further assessed", "By creating a composite based on excess ammonium compared to the five year running average…" etc.

We have rewritten the abstract completely:

"Greenland ice cores provide information about past climate. Few impurity records covering the past two decades exist from Greenland. Here we present results from six firn cores obtained during a 426 km long northern Greenland traverse made in 2015 between the NEEM and the EGRIP deep drilling stations situated on the Western and Eastern side of the Greenland ice sheet, respectively. The cores (9 to 14 m long) are analysed for chemical impurities and cover time spans of 18 to 53 years (±4 yrs) depending on local snow accumulation that decreases from west to east.

The high temporal resolution allows for annual layers and seasons to be resolved. Insoluble dust, ammonium, and calcium concentrations in the 6 firn cores overlap, and also the seasonal cycles are similar in timing and magnitude across sites, while peroxide ($H_2O_2$) varies spatially because it is accumulation dependent and conductivity likely influenced by sea salts, also vary spatially.

Overall, we determine a rather constant dust flux over the period, but in the recent years (1998-2015) we identify an increase in large dust particles that we ascribe to an activation of local Greenland sources. We observe an expected increase in acidity and conductivity in the mid 1970's as a result of anthropogenic emissions followed by a decrease due to mitigation. Several volcanic horizons identified in the conductivity and acidity records can be associated with eruptions in Iceland and in the Barents Sea region. From a composite ammonium record we

obtain a robust forest fire proxy associated primarily with Canadian forest fires (R=0.49). "

**P2L17.** Maybe here a word about possible complications with CFA measurements is appropriate, such as the intrinsic diffusion-like smoothing of the CFA system. **Modified "**CFA represents a highly efficient and rapid analysis technique relative to the measurement of discrete samples, despite its intrinsic dispersion of the signal and small sample loss around core breaks and is favoured for the effective sample decontamination and high sampling resolution (Breton et al., 2012; Erhardt et al., 2022)."

**P2L19-23.** I find the here-stated motivation for the paper a bit vague, e.g., "constraining proxies analysed by means of CFA" could be understood in a technical sense from a measurement quality point of view, which is I guess not what you have in mind. Could you elaborate more precisely on the main aims of the study? We have reiterated the text "We evaluate the impurity concentrations as determined by means of CFA in six shallow Northern Greenland firn cores across Northern Greenland sites. The cores are dated individually to allow comparisons of temporal and spatial trends in both mean concentrations and seasonal cycles. Further we investigate extreme events, such as the deposition from forest fires and volcanic eruptions, and their representation between the 6 sites. The sites chosen cover the lower accumulation area in the central North Greenland, both east and west of the divide, and has only limited prior analysis of this kind (Du et al., 2019a; Vallelonga et al., 2014; Fischer et al., 1998; Gfeller et al., 2014; Schüpbach et al., 2018; Kjær et al., 2021a). "

**P4L6.** I guess by "acid" you refer to the $H^+$ measurements here, which is, however, unclear at this point, since you use one or the other term throughout the text, and it is also a bit misleading, since in normal language acid could mean any kind of acid (I guess you refer to the Brønsted–Lowry acid definition here?). Please choose one terminology, introduce it here and then use it consistently throughout the text. Thanks for pointing this out. We will call it $H^+_{dye}$ when referring to the acid measure in the firn cores using the dye technique, as also done in Kjær et al. 2015 and Winstrup 2019 and acid when referring to volcanic eruptions as that can be many types of acid.

The same goes for the other species, which you alternatingly refer to either by their chemical composition (e.g., $NH_4^+$) or by the common name (i.e., ammonium). The text would be much easier to follow if you sticked to one option throughout. We have changed from the spelled out version to the chemical version as suggested ($NH_4^+$, $Ca^{2+}$ and $H_2O_2$).

**P5L1-11.** This method description is hard to follow and seems incomplete. What I understand you do in essence is seasonal layer counting to derive an age-depth relationship for your cores, for which you use the peroxide mainly, and additionally calcium, if the former has not good enough quality. Indeed, that is the case. We have rewritten the text to clarify.

What remains a bit unclear is how you derive the age scale in Fig. 2; I guess you use the age-depth relationship from the peroxide peaks to interpolate your depth series into a time series using the constant accumulation assumption stated in the second paragraph.

However, this is not entirely clear since you mention the equal accumulation assumption and formal month definition only in relation to "investigating the seasonality" (Fig. 3). We have rewritten the chronology section to make the methodology related to the making of age scale more transparent.

In addition, from the caption of Table 2 it seems that you block-average your depth series data into monthly means following the formal month definition; is that correct? If so, it should be mentioned here. The section have been rewritten.

**P5L6-9.** But could you maybe give an educated guess for how far off you might be with the constant accumulation assumption from the actual seasonal accumulation variations? It is hard to make such an educated guess, as that would require knowledge about daily or seasonal accumulation. This kind of data is available from EastGRIP and NEEM from weather stations, but only for a short time period of a few years. For the remainder of the sites no such weather station data exists. Thus we have chosen not to go into this discussion. Re-analysis data from central Greenland get the annual mean accumulation wrong by factors of ~30%, and thus I would expect the seasonal data to be potentially worse. Thus this approach as also mentioned in the text seems not viable.

That noted, for NEEM Gfeller et al. (2015) states; *"Model evaluations of summer–winter accumulation ratio show consistently more accumulation during summer than during winter (Steen-Larsen et al., 2011). Accumulation data from the Greenland Climate Network (GC-Net) Automated Weather Stations (Steffen et al., 1996) at NEEM and Humboldt (closest automated weather station to NEEM) show significant gaps due to power failure but, nevertheless, they point to a more equal summer to winter ratio than in the model."*

Suggesting our equal accumulation scenario is fair. We have added the following line to highlight that " Accumulation from the GC network at NEEM suggests that a fairly equal summer to winter ratio (Gfeller et al., 2014) and thus we stick with a simple constant accumulation scenario (Gfeller et al., 2014; Kjær et al., 2013). We could have used re-analysis accumulation data to constrain the monthly accumulation, but even high-resolution weather re-analysis performs poorly on the central ice sheet (Kjær et al., 2021c)."

**P5L13.** "profiles": If I understand your methods correctly, Fig. 2 actually shows monthly mean time series for the individual impurity species and cores. This should be explicitly mentioned/repeated here to ease understanding and to avoid confusion with the original depth series. No the data presented in Figure 2 is in full resolution on a timescale (better than monthly). We have changed the word to data

**P7L5.** Do you mean the interannual variability here? Where do I see that the variability is large, and what do you mean with the "concentration variability between sites is masked"? The section has been rewritten. We have further added a whiskers plot (Figure 3) to better illustrate the data and made correlations for all proxies in the suplementary section S3.

"The interannual variations in the individual records are large for all proxies (whiskers in Figure 3).

Spatial concentration gradients (comparing 15-85%) in insoluble dust, $Ca_2^+$, $NH_4^+$, $H^+$, and conductivity are not easy to distinguish because of the interannual variability and the site specific noise. This despite the fact that the firn cores are spanning a distance of 426 km and accumulation is double or more in the northwest (T2015-A1, T2015-A2, T2015-A3, Table 1) compared to the central

north and northeast (T2015-A4, T2015-A5, T2015-A6) (Kjær et al., 2021c). Gfeller et al. 2014 investigated several shallow cores at the NEEM site and reported that annual deposited aerosol concentrations in shallow firn cores can vary strongly over distances of a few meters. The study pointed out that one drill site could be representative for >60% of the variability within a squared area of 100 $m^2$. We add that in Northern central Greenland for distances >100 kilometres apart significant median concentration changes between sites is not resolved beyond seasonal noise for insoluble dust, $Ca^{2+}$, nor $NH_4^+$. This suggest that the dust and $NH_4^+$ are mainly wet deposited in central Northern Greenland, producing similar concentration across all sites and suggest a single source area for each species far enough distant that individual weather events are not influencing the signal. "

"

**P7L19-20.** How do you derive that conclusion, based on the values in Table 2? Did you perform any statistical test to check whether the null hypothesis of identical mean and/or distribution cannot be rejected? Yes, the conclusion was derived based on the values in Table 2 solely. We will add a table of correlation values between records in supplementary and in main text when relevant.

**P8L2-4.** Speaking of spatial variability here is misleading, since variability is more commonly understood to mean random variations. I could imagine what you instead observe here is a spatial gradient in concentration due to a gradient in accumulation. We have rewritten the sentence to clarify "We observe a clear dependence on accumulation in $H_2O_2$ (Table 2)"

**P8L13.** Rather use "average seasonality", "average seasonal cycle" or "climatology" for describing these results. Thank you for these suggestions that we have rephrased the word variability throughout the text to better clarify what we see.

**P10L3-4.** "The variability is high and unevenly distributed" – again, where can I see this? Can you quantify it, i.e., it is high relative to what? What means unevenly distributed?

Sentence have been rewritten.

"The variability is high between the individual years (Figure 2 and Figure 3) and the annual maximum is wide and not very sinusoidal as evidenced in the seasonal cycle of the 15-85% quartiles (Figure 4, top, second).This is a result of an additional source in summer and early autumn namely the Canadian forest fires, and the uneven seasonal shape is evidenced more so in the cores closest to the Canadian forest fire source area (T2015-A1 and T2015-A2)."

The 15-85% quartiles can be seen in Figure 4 and Table S4 and we will add also a whiskers plot (Figure 3) to illustrate it

**P10L14.** It is unclear to which species you refer to here. What means "high deviations in adjacent months"? Isn't that in general the case for a seasonal cycle? We have rewritten the section "Minimum concentrations are found in the summer months July and August. The $Ca^{2+}$ seasonal cycle is smooth compared to that of the insoluble dust, where we observe high insoluble dust loads also in the adjacent months of the annual maximum as evidenced by the monthly 85% quantile (Figure 4, middle left). In the cores T2015-A4, T2015-A5 (EastGRIP site) and T2015-A6 (central divide) it looks like insoluble dust is deposited twice a year (early spring and late autumn/early winter). Whilst this may be due to a local

source as was speculated in other areas of Greenland (Amino et al., 2020; Bullard and Mockford, 2018; Nagatsuka et al., 2021), it could also be ascribed the fact that deposition events are rare in north Central Greenland (McIlhattan et al., 2020) and thus the dust maxima could be found in other formal months."

**Technical comments** (by far not exhaustive)

**OK Throughout text.** The core names are inconsistently labelled either T2015-A1 or 2015TA1, and so on. Please use one consistent nomenclature.

**Ok P1L24.** Change "70's" to "1970s" (more similar instances throughout the text).

**Rephrased P1L25-26.** The sentence "After detrending using…" is difficult to understand and should be rephrased.

**Omitted P2L3.** "intricate": I would avoid such an evaluative adjective in a scientific text.

**Ok P2L11.** Should be changed to "at the deposition site".

**Ok P2L5 and L12.** Please note the hyphenation needed in phrases such as "large-scale atmospheric circulation patterns" or "high-resolution climatic signals". This is a frequent mistake needing correction throughout the text.

**P2L18.** "sample decontamination": To my understanding this means cleaning from toxic components or from radioactive radiation. Is this what you actually mean here? Decontamination here is the removal of the outside ice as it can contain impurities that infer with the analysis. I believe the word decontamination is sufficiently wide and suitable in this context also. Further the word is used in other articles about CFA eg, Kaufmann et al. (2008) and Bigler et al. (2011), Morganti et al. (2007) with more

**OK P2L25.** "Neem" should be "NEEM"; "May to June" of which year do you mean? You also should explain the various site acronyms at some point in the manuscript (preferably at their first respective instance).

**P2L26,27.** Please format the site coordinates correctly. Besides, I would welcome giving the coordinates in decimal degrees, since that is easier to handle in a numerical context.

"Six shallow firn cores were collected during the NEEM to EastGRIP (N2E) traverse in May to June (Karlsson et al., 2020). The traverse went from the NEEM (The North Greenland Eemian Ice Drilling) deep ice core drill site (77.5°N, 51.0°W, 2481 m a.s.l.) to the EastGRIP (The East Greenland Ice-core Project) deep ice core drill site (75.64°N, 36°W, 2712 m a.s.l.)."

**OK P2L31.** A comma is missing between "Greenland" and "and then shipped".

**OK P3L2.** Please rephrase to "prior to the CFA measurements".

**OK Fig. 1 caption.** Please mention the information relevant to the study first, i.e., first the firn cores, then afterwards the information on the surface elevation data set.

**Rephrased Table 1 caption.** The column of core depths is not mentioned in the caption. Additionally, you write that the core labels go till "2015T-A5", but there is another core ("2015T-A6") listed in the table.

**Rephrased P4L4.** "in 2017": this could be mistaken to mean that you only measured the impurity content for the year 2017; I guess instead you mean the CFA measurements took place in 2017; please rephrase.

**Ok P4L5.** Add a comma before "by adding".

**Ok P4L8.** Please change to "were converted into units of concentration".

Rephrased **P4L9-10.**

- Do you mean "A baseline was established"?
- You should explain what "milliq water" means; not every reader might be familiar with the laboratory terminologies.
- What is "8eight 55 cm pieces stacked" supposed to mean?
- "Although" is not the correct wording here; I guess you mean something around "In general the baseline was established by… However, for the top 1.65 metres, where the core was fragile […], the baseline was established…". Please clarify.

**Rephrased P4L13/15.** The firn cannot "suck anything". Melt water can flow or percolate into the firn driven by capillary forces; please use the correct physical terminology.

**Rephrased P4L15.** I don't understand how excess water can be limited to an amount of 0.5-1 cm; what does this unit mean here? Please bear in mind that not every reader might have worked with a CFA system him- or herself.

**P4L18.** "response time": Again, a reader not familiar with the CFA technique will have problems understanding this; what do you mean by response time and how does this affect the effective depth resolution?

The term is commonly used e.g. Emanuelsson et al. (2015), Maselli et al. (2013), but we have rewritten the text to make it easier for the unfamiliar reader "while for the conductivity and dust with shorter step-change response times (time it takes to go from a level of 5% to 95% of a concentration) a depth resolution of 8 mm was achieved"

**Ok P4L20.** Please change to "at a sufficient resolution".

**Rephrased P4L21.** Do you mean "which are used to constrain…"?

**Ok P4L22-22.** Please change to "as it is produced by a photochemically-derived".

**OK P4L27.** Please change to "this exchange can cause smoothing".

**OK P5L3.** Please change "invoked" to "used". . **P5L9-11.** Why not? If you mention this explicitly here then you should give a reason for not doing it.

We believe we do give a reason namely that it is known that "even high-resolution weather re-analysis performs poorly on the central ice sheet". We however mention it because splitting the accumulation into seasons is sometimes

done in other studies (eg gfeller), mainly when direct evidence of precipitation from weather stations have been obtained for several years. As this is not the case for all of our sites we refrain from splitting the year into a fancier precipitation scenario

**Figure 3.** It is rather counterintuitive to display the formal months in the reversed temporal order summer – spring – winter – autumn. We will reverse the order

**OK P20 L27.** "dissolves" is the wrong wording, please use "resolves" instead.

**P20L29.** As mentioned earlier, you mix up spatial variability (random variations) with spatial gradients or spatial variations. Please be careful to use the appropriate wording throughout the text. We have gone through the text and rewritten accordingly

**Rephrased P21L4.** Please change to "We thus highlight" and to "of using the same methods".

**OK P21L6.** Please change to "in the acid and conductivity profiles".

**Ok P21L7.** Please change to "an increase over time, especially for the large …".

**Rephrased P21L9-11.** Please change the reference to the standard format; the final sentence is grammatically wrong.

Powered by TCPDF (www.tcpdf.org)

---

## Author Response (AR2)

In addition to the below comments from the reviewer which have been modified. Further additional grammatical changes have been conducted and Figure6 and Figure 1 have been updated. Figure 6 as it was noted that the horizontal line colors had been swapped between acid and conductivity. Figure 1 to not contain a white bar as noted by the reviewer.

On behalf of the authors

Helle Kjær

General comments

The Authors accomplished a substantial work of revision of the manuscript upon the suggestions of the referees, amending ample parts of the texts and adding/improving many important figures and tables. The achieved results appear to be better supported now, by figures, tables, and references and the main objectives of the paper are much clearer.
I do appreciate the efforts of the Authors; my concerns and questions were addressed quite satisfactorily and to me the paper is now close to an acceptable version.

We are also very pleased with the how you and the other reviewers' great suggestions contributed to the current version of the paper

However, I find that a second careful revision would be useful to complete the work, especially concerning the English language (grammar, wording, orthography).
I acknowledge the work of the Authors in improving the English level, being myself a non-native English speaker as well, but I think that another round could be useful to fix some mistakes throughout the text.

The English co-authors have gone through the paper and additional modifications were implemented throughout. In addition to your specific comments, which were implemented as suggested.

Here below I am listing some specific comments about English language and a few other comments on the point-to-point reply and on the revised version.

Specific comments (pages and lines are referred to the version with tracked changes)

Line 25 page 1. I would add the verb, i.e., "conductivity is likely…" and would use the singular for "sea salt".

Rewritten in the following way "while peroxide ($H_2O_2$) and conductivity both have spatial variations. $H_2O_2$ driven by the accumulation pattern and conductivity is likely influenced by sea salt"

ine 21 page 2. The sentence does not flow smoothly, I feel. Maybe it is better to make it explicit? For instance, "…in the ice, which are important…"
Changed accordingly

Line 11 page 3. I cannot understand what "has limited prior analysis". Do the Authors refer to the investigated area? It is not perfectly clear; subject appears to be missing.

The sentence know reads "The sites chosen represent cover the lower accumulation area in the 10 central North Greenland, both east and west of the divide, and has only limited prior analysis of this kind

"

Figure 1 page 3. In both the versions of the paper (with and without tracked changes) a white rectangular spot appears in the figure.

Figure has been modified

Line 14 page 7. Just a comma would be fine after "resolved"

Changed accordingly.
Line 24 page 7. "That" should be removed.

Changed accordingly
Lines 15-17 page 12. Check grammar in this sentence.

Line 23 page 12. I would use "sea salt" instead of "sea salts", as written earlier. Please, change it throughout the text.

Changed throughout
Line 26 page 12. "anthropogenic changes…are", not "is".

Changed accordingly
Line 11 page 22. I doubt that "very influenced" is correct in English.

Changed to significantly

Supplement material, Table S1 page 2. A typo is present in "Multi-function…"

Changed accordingly

Please, check punctuation after the changes.

My comment for lines 2-4 page 8 (first version). My concern was exactly about the description of different variability between Western and Eastern sites, mainly. Now, thanks also to the correlation study and whiskers plot the analysis of variability is much clearer.

We are happy that it is now clear

My comment for line 20 page 7 (first version). I gather here that the Authors mean #particles/mL, which is fine but through the text it is referred to as dust flux or concentration and it would be clearer by using one of the two expressions only.

In Figure 3 we still use counts/mL as that is the original measurement. The fluxes are subject to larger uncertainty as they depend also on the accumulation and the uncertainty within. Thus we stick with both expressions.